# Geometry of antiparallel microtubule bundles regulates relative sliding and stalling by PRC1 and Kif4A

Sithara Wijeratne[1,2], Radhika Subramanian[1,2]*

[1]Department of Molecular Biology, Massachusetts General Hospital, Boston, United States; [2]Department of Genetics, Harvard Medical School, Boston, United States

**Abstract** Motor and non-motor crosslinking proteins play critical roles in determining the size and stability of microtubule-based architectures. Currently, we have a limited understanding of how geometrical properties of microtubule arrays, in turn, regulate the output of crosslinking proteins. Here we investigate this problem in the context of microtubule sliding by two interacting proteins: the non-motor crosslinker PRC1 and the kinesin Kif4A. The collective activity of PRC1 and Kif4A also results in their accumulation at microtubule plus-ends ('end-tag'). Sliding stalls when the end-tags on antiparallel microtubules collide, forming a stable overlap. Interestingly, we find that structural properties of the initial array regulate microtubule organization by PRC1-Kif4A. First, sliding velocity scales with initial microtubule-overlap length. Second, the width of the final overlap scales with microtubule lengths. Our analyses reveal how micron-scale geometrical features of antiparallel microtubules can regulate the activity of nanometer-sized proteins to define the structure and mechanics of microtubule-based architectures.

DOI: https://doi.org/10.7554/eLife.32595.001

## Introduction

The organization of microtubules into specialized architectures is required for a diverse range of cellular processes such as cell division, growth and migration (*Dogterom and Surrey, 2013*; *Subramanian and Kapoor, 2012*). Microtubule-crosslinking proteins play important roles in determining the relative orientation, size and dynamics of microtubule-based structures. These proteins include molecular motors that utilize the energy from ATP hydrolysis to mediate the transport of one microtubule over another (referred to as 'relative sliding') (*Sharp et al., 2000*; *Tolić-Nørrelykke, 2008*; *Forth and Kapoor, 2017*). Motor proteins frequently act in conjunction with non-motor microtubule crosslinking proteins that oppose relative sliding and regulate both the stability and the size of the arrays (*Dogterom and Surrey, 2013*; *Subramanian and Kapoor, 2012*; *Bratman and Chang, 2008*). The activities of motor and non-motor proteins are in turn modulated by the microtubule cytoskeleton. At the nanometer length-scale, numerous tubulin isotypes and post-translational modifications on tubulin act as a code to tune the activity of microtubule-associated proteins (MAPs) (*Gull et al., 1986*; *Ludueña, 2013*; *Yu et al., 2015*; *Gadadhar et al., 2017*). In addition, it is becoming apparent that at the micron length-scale, the geometrical properties of microtubule bundles, such as orientation, filament length and overlap length, also modulate the output of motor and non-motor proteins (*Fink et al., 2009*; *Kuan and Betterton, 2016*; *Shimamoto et al., 2015*; *Braun et al., 2017*). Currently, we have a limited understanding of the mechanisms by which the micron-sized features of a microtubule network are 'read' and 'translated' by associated proteins.

Arrays of overlapping antiparallel microtubules form the structural backbone of diverse cellular structures. Several insights into the mechanisms underlying the assembly of such arrays have come from examining the non-motor antiparallel microtubule crosslinking proteins of the PRC1/Ase1/

*For correspondence:
radhika@molbio.mgh.harvard.edu

**Competing interests:** The authors declare that no competing interests exist.

MAP65 family. These evolutionarily conserved proteins play an important role in organizing microtubule arrays in interphase yeast and plant cells, and subsets of spindle microtubules in dividing cells in all eukaryotes (*Chan et al., 1999*; *Loïodice et al., 2005*; *Yamashita et al., 2005*; *Jiang et al., 1998*; *Mollinari et al., 2002*; *Polak et al., 2017*). These passive non-motor proteins act in concert with a number of different motor proteins, such as those of the kinesin-4, kinesin-5, kinesin-6 and kinesin-14 families (*Jiang et al., 1998*; *Mollinari et al., 2002*; *Zhu et al., 2006*; *Gruneberg et al., 2006*; *Janson et al., 2007*; *D'Avino et al., 2007*; *Fu et al., 2009*; *Bieling et al., 2010*; *Braun et al., 2011*; *Duellberg et al., 2013*; *Subramanian et al., 2013*; *Pringle et al., 2013*; *de Keijzer et al., 2017*). A subset of these kinesins, such as Kif4A, Kif23 and Kif20, directly or indirectly bind PRC1/MAP65/Ase1 family proteins (*Gruneberg et al., 2006*; *Fu et al., 2009*; *Bieling et al., 2010*; *Subramanian et al., 2013*; *Kurasawa et al., 2004*; *Zhu and Jiang, 2005*; *Vitre et al., 2014*). The diversity in the properties of motor proteins that act in conjunction with the different PRC1 homologs affords a powerful model system to elucidate the biophysical principles governing the organization of antiparallel microtubule arrays. However, thus far, the mechanistic studies of PRC1-kinesin systems have mainly focused on elucidating how microtubule sliding by kinesins is regulated by PRC1 homologs (*Braun et al., 2011*; *Subramanian et al., 2010*; *Lansky et al., 2015*). How the geometry of PRC1-crosslinked microtubules, such as lengths of microtubules and the size of the initial overlap, modulates the activities of associated motor proteins is poorly understood.

Here, we address this question by examining the relative sliding of PRC1-crosslinked antiparallel microtubules by the kinesin Kif4A. The collective activity of PRC1 and Kif4A is required for the organization of the spindle midzone, an antiparallel bundle of microtubules that is assembled between the segregating chromosomes at anaphase in dividing cells (*Kurasawa et al., 2004*; *Zhu and Jiang, 2005*; *Shrestha et al., 2012*; *Nunes Bastos et al., 2013*; *Hu et al., 2011*). Kif4A, a microtubule plus-end directed motor protein is recruited to the midzone array through direct binding with PRC1, where it acts to suppress microtubule dynamics (*Bieling et al., 2010*; *Subramanian et al., 2013*; *Nunes Bastos et al., 2013*; *Hu et al., 2011*). Previous in vitro studies with the *Xenopus Laevis* homologs of these proteins also suggest that they can drive the relative sliding of antiparallel microtubules over short distances (*Bieling et al., 2010*). However, microtubule sliding by Kif4A and its modulation by the geometrical features of the initial PRC1-crosslinked microtubules remain poorly characterized. In addition to sliding, the processive movement of PRC1-Kif4A complexes and their slow dissociation from the microtubule end result in the accumulation of both proteins in micron-sized zones at the plus-ends of single microtubules (hereafter referred to as 'end-tags'). It is observed that: (i) the velocity of the motor movement is reduced at end-tags (*Subramanian et al., 2013*). This hindrance to motor stepping is likely due to molecular crowding at microtubule ends (*Subramanian et al., 2013*; *Leduc et al., 2012*). (ii) The size of end-tags increases with microtubule length (*Subramanian et al., 2013*). How the length-dependent accumulation of PRC1-Kif4A molecules on single microtubules impacts the organization of antiparallel bundles is unknown.

Here, we show using TIRF-microscopy based assays that the collective activity of PRC1 and Kif4A results in relative microtubule sliding and concurrent end-tag formation on antiparallel microtubules. Interestingly, we find that PRC1-Kif4A end-tags act as roadblocks to prevent the complete separation of sliding microtubules. Consequently, sliding and stalling of antiparallel microtubules by PRC1 and Kif4A result in the assembly of a stable overlap that is spatially restricted to the filament plus-ends. Surprisingly, quantitative examination of the data reveals that two aspects of the PRC1-Kif4A-mediated microtubule organization are modulated by the initial geometry of crosslinked microtubules. First, the sliding velocity in this system scales with the initial length of the antiparallel overlap. Second, the size of the final stable antiparallel overlap established by PRC1 and Kif4A scales with the lengths of the crosslinked microtubules. Our observations provide insights into the principles by which the geometrical features of antiparallel arrays can be translated to graded mechanical and structural outputs by microtubule-associated motor and non-motor proteins.

## Results

### Collision of PRC1-Kif4A end-tags on sliding microtubules results in the formation of antiparallel overlaps of constant length at steady-state

To investigate microtubule sliding in the PRC1-Kif4A system, we reconstituted the activity of the kinesin Kif4A on a pair of antiparallel microtubules crosslinked by the non-motor protein PRC1. For these studies, we adapted a Total Internal Reflection Fluorescence (TIRF) microscopy-based assay that we have previously used to examine relative sliding of PRC1-crosslinked microtubules by the motor-protein Eg5 (*Subramanian et al., 2010*). First, biotinylated taxol-stabilized microtubules, labeled with rhodamine, were immobilized on a glass coverslip. Next, unlabeled PRC1 (0.2 nM) was added to the flow chamber and allowed to bind the immobilized microtubules. Finally, rhodamine-labeled non-biotinylated microtubules were flowed into the chamber to generate microtubule 'sandwiches' crosslinked by PRC1 on the glass coverslip (*Figure 1A*). After washing out the unbound proteins, the final assay buffer containing Kif4A-GFP, PRC1 and ATP at specified concentrations was flowed into the chamber to initiate end-tag formation and microtubule sliding (*Figure 1A*). Near-simultaneous multi-wavelength imaging of rhodamine-labeled microtubules and Kif4A-GFP showed that Kif4A preferentially accumulates in the overlap region of PRC1-crosslinked microtubules (*Figure 1B–D*; $t$ = 0 s; 0.2 nM PRC1 + 6 nM Kif4A-GFP). This is in agreement with prior findings that PRC1 selectively accumulates at regions of antiparallel microtubule overlap regions and recruits Kif4A to these sites (*Bieling et al., 2010*; *Subramanian et al., 2010*). In the example shown in *Figure 1B-D*, the average fluorescence intensity of Kif4A-GFP in the microtubule overlap region is 5-fold higher than the fluorescence intensity in the non-overlapped region at the first time point recorded (*Figure 1B–E*; $t$ = 0 s). In addition, time-lapse imaging shows an enhanced accumulation of Kif4A-GFP at the plus-ends of both the crosslinked microtubules. We refer to this region of high protein density at microtubule plus-ends as 'end-tags' (*Figure 1B–E*; $t$ = 10–40 s; ~2.5 fold enrichment of Kif4A-GFP at end-tags over the untagged overlap at 10 s). Therefore, under these experimental conditions Kif4A-GFP-containing end-tags are established at the plus-ends of crosslinked microtubules.

Time-lapse imaging and kymography-based analyses revealed that the end-tagged antiparallel microtubules slide relative to each other (*Figure 1B–D and F–H*). Strikingly, we find that microtubule sliding stalls when the end-tags arrive at close proximity (*Figure 1B–D and F–H*). This results in the formation of stable antiparallel overlaps that maintain a constant steady-state width for the entire duration of the experiment (*Figure 1B–D and F–H*; $t$ = 10 min). We rarely (5%) observe sliding microtubules stall before they arrive at the plus-end of the immobilized microtubule, and these occasional premature stalling events may arise from non-specific sticking to the glass coverslip. Under these experimental conditions, we do not observe any event where the moving microtubule slides past the end-tag of the immobilized microtubule. These observations indicate that the formation of stable antiparallel overlaps is due to the end-tags on the crosslinked microtubule pair arriving at close proximity during relative sliding.

We next examined PRC1 localization on sliding microtubules by conducting experiments similar to that described above, except with GFP-labeled PRC1 and unlabeled Kif4A (*Figure 1I–K*; 0.5 nM GFP-PRC1 + 6 nM Kif4A). We find that the localization pattern of GFP-PRC1 is similar to Kif4A with the highest fluorescence intensity at the end-tags, intermediate intensity at the untagged microtubule overlap regions and the lowest intensity on single microtubules. Similar to the observations in *Figure 1F–H*, we find that sliding microtubules stall when their end-tags arrive in close proximity (*Figure 1I–K*).

Together, these observations suggest that human PRC1-Kif4A complexes can drive the relative sliding of antiparallel microtubules over the distance of several microns. However, sliding comes to a halt at microtubule plus-ends resulting in the formation of stable antiparallel overlaps of constant steady-state length.

### Molecular determinants of sliding and cross-bridging in the PRC1-Kif4A system

To investigate the molecular determinants of the observed sliding and cross-bridging in the PRC1-Kif4A system, we examined if Kif4A alone can crosslink microtubules. A common mechanism by

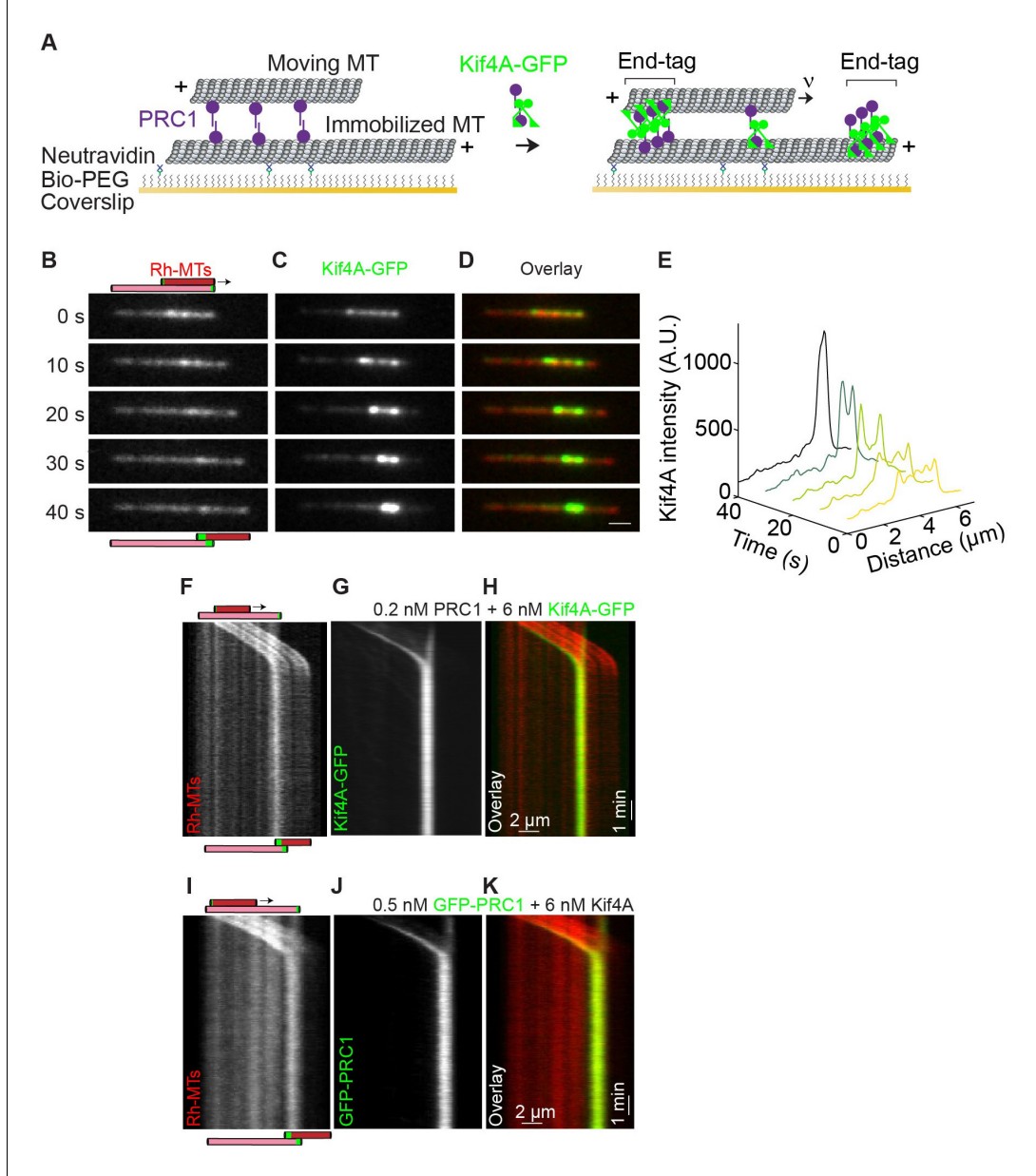

**Figure 1.** Relative microtubule sliding and the formation of stable antiparallel microtubule overlaps by PRC1 and Kif4A. (A) Schematic of the in vitro assay. A biotinylated microtubule ('immobilized MT', X-rhodamine labeled) immobilized on a PEG coated coverslip and a non-biotinylated microtubule ('moving MT', X-rhodamine-labeled) are crosslinked in an antiparallel orientation by PRC1 (purple). Microtubule sliding and end-tag formation are initiated by addition of Kif4A-GFP (green), PRC1 and ATP. (B–D) Representative time-lapse fluorescence micrographs of relative microtubule sliding in experiments with 0.2 nM PRC1 and 6 nM Kif4A-GFP. Images show (B) a pair of X-rhodamine-labeled microtubules, (C) Kif4A-GFP, and (D) overlay images (red, microtubules; green, Kif4A-GFP). The schematic in (B) illustrates the position and relative orientation of both the immobilized (pink) and moving (red) microtubules and the end-tags (green) at the beginning and end of the time sequence. Scale bar: $x$: 2 µm. (E) Line scan analysis of the Kif4A intensity from the micrographs in (C) shows the distribution of Kif4A within the overlap at the indicated time points. (F–H) Kymographs show the relative sliding and stalling of antiparallel microtubules (F), associated Kif4A-GFP (G) and the overlay image (red, microtubules; green, Kif4A-GFP) (H). Assay condition: 0.2 nM PRC1 and 6 nM Kif4A-GFP. Scale bar: $x$: 2 µm and $y$: 1 min. (I–K) Kymographs show the relative sliding and stalling of antiparallel microtubules (I), associated GFP-PRC1 (J) and the overlay image (red, microtubules; green, GFP-PRC1) (K). Assay condition: 0.5 nM GFP-PRC1 and 6 nM Kif4A. Scale bar: $x$: 2 µm and $y$: 1 min.

DOI: https://doi.org/10.7554/eLife.32595.002

which dimeric kinesins crosslink and slide microtubules is by interacting with one microtubule via the motor domains and another microtubule using non-motor C-terminus domains. Whether the C-terminus of human Kif4A, which binds both DNA and PRC1, can also bind microtubules is unknown (*Subramanian et al., 2013*). To examine this, we purified the C-terminus PRC1 and DNA-binding domain of Kif4A (aa. 733–1232) and performed microtubule co-sedimentation assays (*Figure 2A*). Dose-dependent microtubule binding of this domain was not observed in the tubulin concentration range tested. Therefore, in the cross-bridging complex, the C-terminus of Kif4A is likely to associate with the N-terminus of PRC1, which forms the link between both microtubules (*Figure 2B*).

Next, we examined if there was a non-canonical mode of microtubule sliding by Kif4A in the absence of PRC1. For this, we attempted microtubule crosslinking experiments as described in *Figure 1* with Kif4A alone (2 mM ATP). Similar to a previous report with Xklp1, full-length human Kif4A does not bundle microtubules in the presence of ATP (not shown) (*Bieling et al., 2010*). We reasoned that bundle formation might be enhanced if the lifetime of the protein on microtubule was increased. To investigate this, we first bound Kif4A-GFP to immobilized microtubules at low ATP concentrations (6 nM Kif4A-GFP + 10 nM ATP). Under these conditions, due to the slow rate of stepping, the protein is bound along the entire length of the microtubules (*Figure 2C and D*). Subsequent addition of non-biotinylated microtubules resulted in the formation of pairs of crosslinked microtubules (~20% microtubules are crosslinked per 133 μm$^2$ field of view under our experimental conditions). We find that the predominant angle of initial attachment between the two microtubules is 0–30° (*Figure 2E*). Finally, we flowed in 2 mM ATP and 6 nM Kif4A-GFP, which initiated end-tag formation on all microtubules. We find that contact between the end-tag on the non-biotinylated microtubule with the immobilized microtubule results in tip-mediated movement of one microtubule over the other (*Figure 2C–D*). Similar to experiments with PRC1 and Kif4A, sliding completely stalls when the Kif4A-GFP end-tags collide. These findings indicate that a Kif4A-dense microtubule tip can slide along another microtubule even though the C-terminus of Kif4A does not bind microtubules with high affinity.

Together these findings suggest that there are two possible modes of cross-bridging and sliding in the PRC1-Kif4A system. First, Kif4A-molecules interacting with microtubule-crosslinking PRC1-molecules can drive sliding. Second, Kif4A molecules at the tips of one microtubule can bind and slide over another microtubule. Currently, we do not know if the second mode of movement occurs in the context of an antiparallel bundle, where the angle between crosslinked microtubules is 180°. Finally, these experiments show that relative microtubule movement can stall at microtubule ends in the absence of PRC1, suggesting that molecular crowding is likely to be the predominant cause of stable overlap formation in the PRC1-Kif4A system.

## Characterization of relative microtubule sliding in the PRC1-Kif4A system

To further characterize relative sliding in mixtures of PRC1 and Kif4A, we quantitatively examined the microtubule movement observed in these experiments. Analysis of the instantaneous velocity during microtubule sliding (*Figure 3A–C*; 0.2 nM PRC1 + 6 nM Kif4A-GFP), reveals three phases: (1) initial sliding at constant velocity, (2) reduction in sliding velocity as the end-tags arrive at close proximity, and (3) microtubule stalling and the formation of stable overlaps that persist for the duration of the experiment.

We first focused on microtubule movement in phase-1 and investigated how the relative solution concentrations of the motor and the non-motor protein impact the initial sliding velocity. This is particularly interesting in the case of PRC1-Kif4A system as the recruitment of Kif4A to microtubules is dependent on PRC1 (*Bieling et al., 2010*; *Subramanian et al., 2013*). Therefore, one possible outcome is that motor-protein movement is sterically hindered at higher PRC1 concentrations resulting in lower sliding velocities. Alternatively, it is possible that more Kif4A is recruited to microtubule overlaps at higher PRC1 concentrations and this could counter the potentially inhibitory effects of PRC1. To distinguish between these mechanisms, we compared the maximum microtubule sliding velocity (computed as the average velocity from phase-1) at two different PRC1:Kif4A concentration ratios (*Figure 3D*). We found that increasing the PRC1 solution concentration 5-fold (0.2 and 1 nM) at constant Kif4A-GFP concentration (6 nM) resulted in a 4-fold reduction in the microtubule sliding velocity (velocity = 46 ± 17 nm/s at 0.2 nM and velocity = 11 ± 8 nm/s at 1 nM). Similarly, in assays with GFP-PRC1 and unlabeled Kif4A, we found that increasing PRC1 concentration 2-fold (0.5 and 1

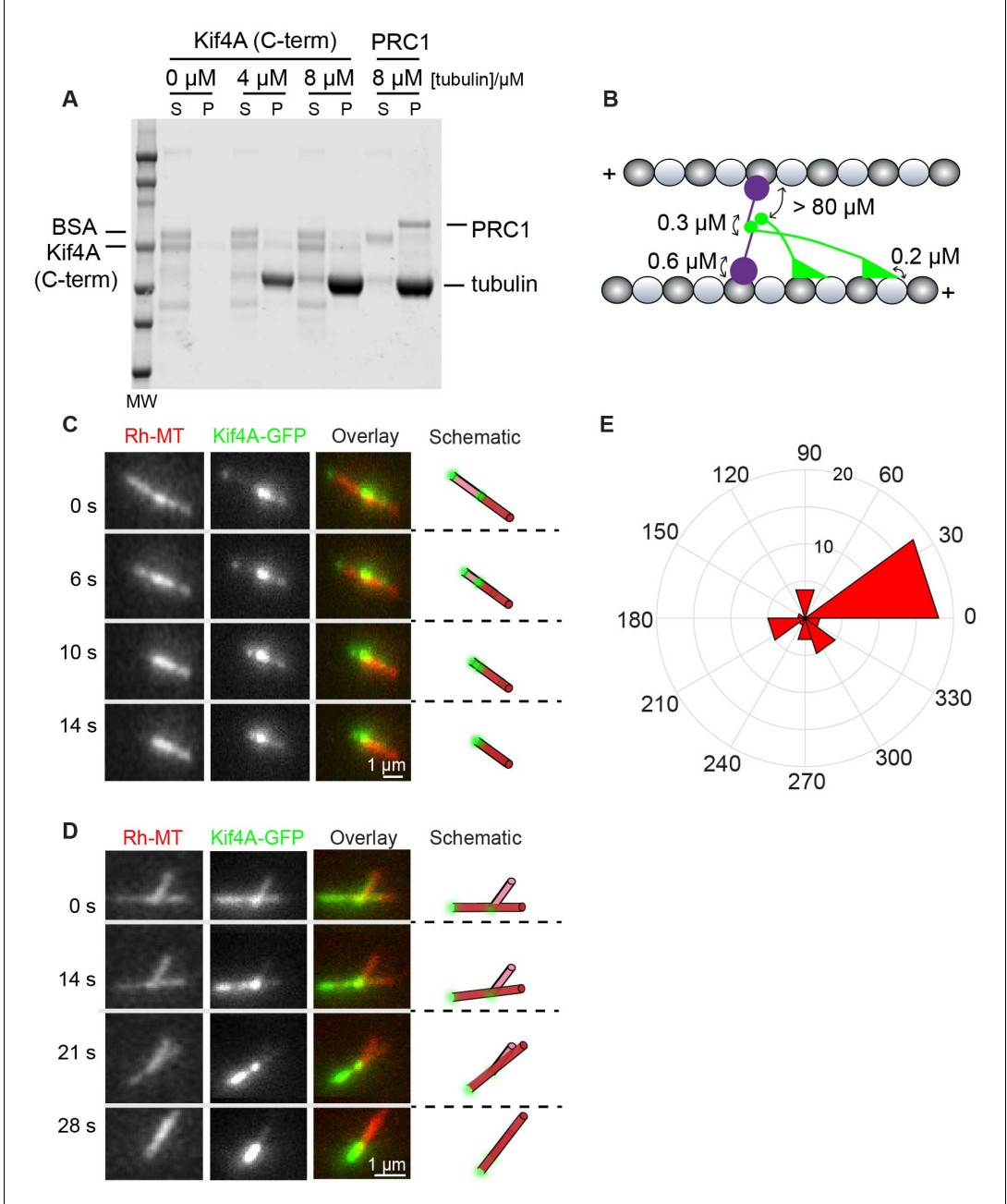

**Figure 2.** Molecular determinants of cross-bridging and sliding in the PRC1-Kif4A system. (A) Microtubule co-sedimentation assay. SDS Page analysis of the interaction of Kif4A (C-term) with increasing amounts microtubules (0–8 μM). Full-length PRC1 was included as a control. Quantification of band intensities: Kif4A_pellet = 8% and 6% at 4 μM and 8 μM tubulin. BSA_pellet = 1% and 3% at 4 μM and 8 μM tubulin. (B) Schematic shows the proposed molecular configuration of the cross-bridging PRC1-Kif4A complex in a microtubule overlap. Known dissociation constants are indicated. (C–D) Representative time-lapse fluorescence micrographs of relative microtubule sliding in experiments with 6 nM Kif4A-GFP + 2 mM ATP. Images show a pair of X-rhodamine-labeled microtubules, Kif4A-GFP, and overlay images (red, microtubules; green, Kif4A-GFP). The schematic illustrates the position and relative orientation of both the immobilized and moving microtubules (red) and the end-tags (green) at the beginning and end of the time sequence. (E) Rose diagram of the initial angle of attachment of the sliding microtubule. The plot shows the most probable angle of attachment is between 0–30° (N = 39). Assay condition: 6 nM Kif4A-GFP + 2 mM ATP.
DOI: https://doi.org/10.7554/eLife.32595.003

The following source data is available for figure 2:

**Source data 1.** This spreadsheet contains the initial angle of attachment of the sliding microtubule used to generate the rose diagram shown in *Figure 2E*.
DOI: https://doi.org/10.7554/eLife.32595.004

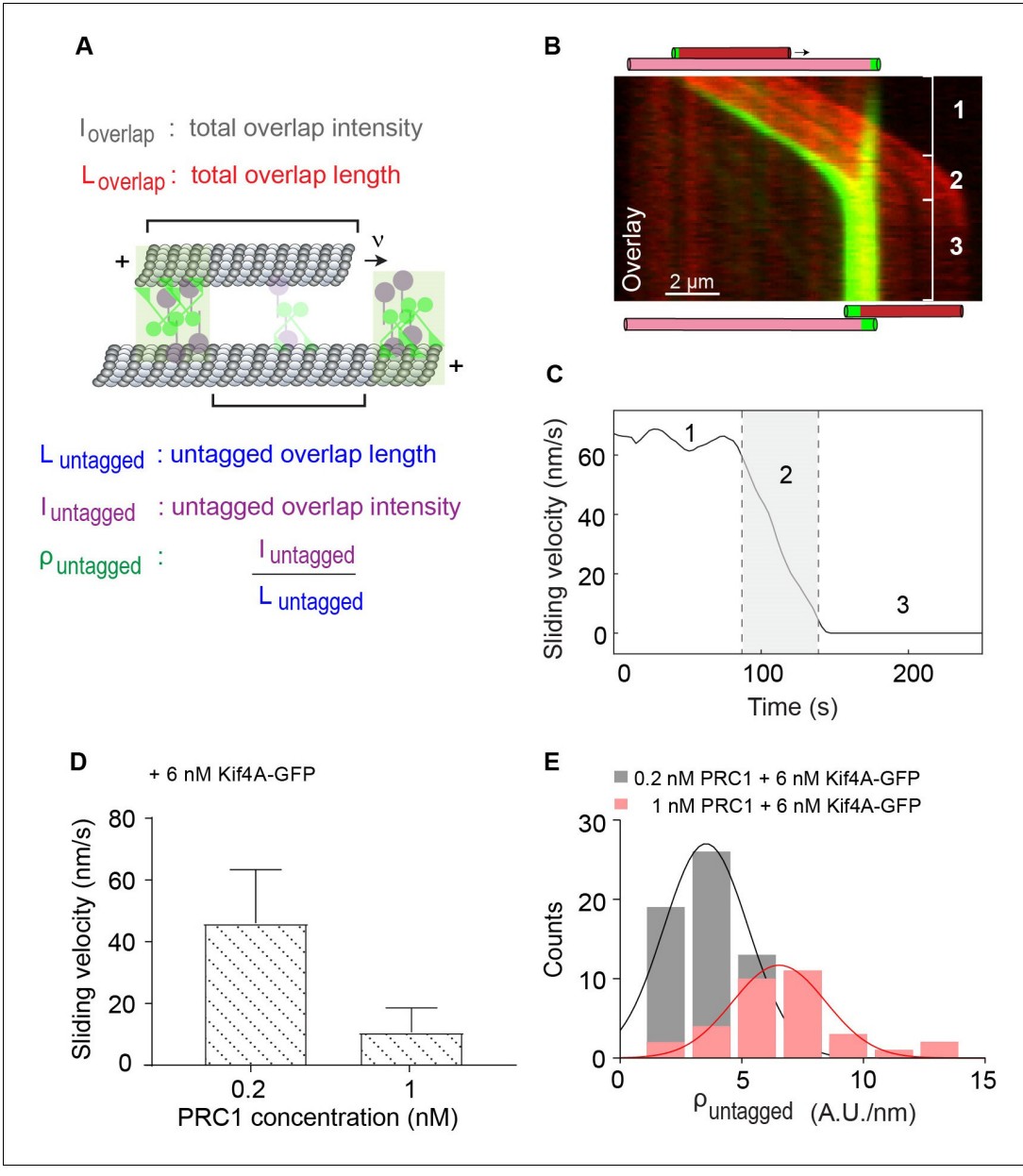

**Figure 3.** Quantitative analysis of microtubule sliding in the PRC1-Kif4A system. (**A**) Schematic of a pair of crosslinked microtubules showing the parameters described in *Figures 1* and *4*. (**B**) Kymograph shows the relative sliding and stalling in a pair of antiparallel microtubules (red) and associated Kif4A-GFP (green). Assay condition: 0.2 nM PRC1 and 6 nM Kif4A-GFP. Scale bar: 2 µm. The schematic illustrates the position and relative orientation of both the immobilized (pink) and moving (red) microtubules and the end-tags (green) at the beginning and end of the time sequence. (**C**) Time record of the instantaneous sliding velocity of the moving microtubule derived from the kymograph in (**B**). The dashed lines demarcate the three phases observed in the sliding velocity profile: (1) constant phase, (2) slow down and (3) stalling. (**D**) Bar graph of the average sliding velocity calculated from the constant velocity movement in phase-1. Assay conditions: (i) 0.2 nM and 6 nM Kif4A-GFP (mean: 46 ± 17; N = 98) (ii) 1 nM PRC1 and 6 nM Kif4A-GFP (mean: 11 ± 8; N = 45). Error bars represent the standard deviation of the data. (**E**) Histograms of the initial GFP-fluorescence density in the untagged region of the overlap, $\rho_{untagged}$. Assay conditions: (i) 0.2 nM PRC1 and 6 nM Kif4A-GFP (black; mean: 3.5 ± 1.7 A.U./nm; N = 64) and (ii) 1 nM PRC1 and 6 nM Kif4A-GFP (red; mean: 6.5 ± 1.9 A.U./nm; N = 33). The mean and error values were obtained by fitting the histograms to a Gaussian distribution.

DOI: https://doi.org/10.7554/eLife.32595.005

The following source data and figure supplements are available for figure 3:

*Figure 3 continued on next page*

*Figure 3 continued*

**Source data 1.** This spreadsheet contains the average sliding velocity used to generate the bar graph shown in Figure 3D and the initial GFP-fluorescence density, $\rho_{untagged}$, used to generate *Figure 3E*.
DOI: https://doi.org/10.7554/eLife.32595.016
**Figure supplement 1.** Average sliding velocity of antiparallel microtubule overlaps by PRC1 and Kif4A.
DOI: https://doi.org/10.7554/eLife.32595.015
**Figure supplement 1—source data 1.** This spreadsheet contains the average sliding velocity used to generate the bar graph.
DOI: https://doi.org/10.7554/eLife.32595.017

nM) at constant Kif4A levels (6 nM) resulted in a ~2 fold reduction in the microtubule sliding velocity (*Figure 3—figure supplement 1*; velocity = 30 ± 13 nm/s at 0.5 nM and velocity = 19 ± 8 nm/s at 1 nM PRC1). Interestingly, in these experiments, we could restore the sliding velocity by compensating the 2-fold increase in the PRC1 concentration with a 2-fold increase in the Kif4A concentration (*Figure 3—figure supplement 1*).

A possible explanation for the reduced velocity at the higher PRC1:Kif4A concentration is that there are fewer Kif4A molecules in the overlap due to competition from PRC1 for binding sites on the microtubule surface. Therefore, we compared the Kif4A-GFP density in the untagged overlap at two different solution PRC1 concentrations (*Figure 3E*). The data show that a 5-fold increase in the PRC1 concentrations results in a 2-fold increase in the average Kif4A density in the untagged overlap region, indicating that Kif4A is effectively recruited to antiparallel overlaps at the highest PRC1 concentrations in our assays (*Figure 3E*). Together, these results are consistent with a mechanism in which the solution concentration ratios of PRC1:Kif4A sets the sliding velocity by determining the relative ratio of sliding-competent PRC1-Kif4A complexes to sliding-inhibiting PRC1 molecules in the antiparallel overlap. The inhibition can arise either due to increased frictional forces or steric inhibition to stepping in crowded overlaps at higher PRC1 concentrations. We elaborate on these possibilities in the discussion section.

## Microtubule sliding velocity in the PRC1-Kif4A system scales with initial overlap length

We next examined if the initial width of the PRC1-crosslinked anti-parallel overlap impacts the sliding velocity in phase-1. The initial width is defined as the overlap length at the first time point imaged after flowing in the final assay buffer in our experiment ($t = 0$; example: first panel in *Figure 1B*). Remarkably, analysis of three different datasets suggests that antiparallel microtubules with longer initial overlaps slide at a higher velocity than microtubules with shorter initial overlaps under the same experimental condition (*Figure 4A–B*). Note: no obvious trend was observed at the higher PRC1 concentration, possibly due to the scatter in the data relative to the low magnitude of sliding velocities (*Figure 4A*; red squares).

Are molecules in the untagged or end-tagged region responsible for the observed overlap length-dependent sliding? To answer this question, we first analyzed the data from the PRC1-Kif4A experiments to determine if the phase-1 microtubule sliding velocity depended on the amount of protein at the end-tags. However, no significant correlation was observed between the sum of end-tag lengths or sum of end-tag intensities and sliding velocity (*Figure 4—figure supplement 1A–B*). Consistent with this, the average sliding velocity (75 ± 25 nm/s; N = 25) is independent of end-tag intensity in experiments with Kif4A alone (*Figure 2C–D*) (*Figure 4—figure supplement 1Q*). Instead, we find that in experiments with PRC1 and Kif4A, the sliding velocity increases with the initial untagged overlap length under all the conditions where length-dependent sliding is observed (*Figure 4—figure supplement 1D–E*). Under these conditions, Kif4A-GFP intensity is proportional to both the total and untagged overlap lengths, suggesting that sliding velocity scales with the number of molecules in the overlap (*Figure 4—figure supplement 1C–D* inset; Note: intensity is a measure of the total number of molecules in the overlap, it is not a direct measure of the number of Kif4A motors involved in binding PRC1 and sliding. We assume that the number of sliding-competent motors is proportional to total intensity). In addition, no correlation is found between sliding velocity and the average motor density in the untagged overlap during phase-1 movement (0.2 nM PRC1 and 6 nM Kif4A-GFP; Pearson's coefficient: −0.15; N = 20) (*Figure 4—figure supplement 1F*).

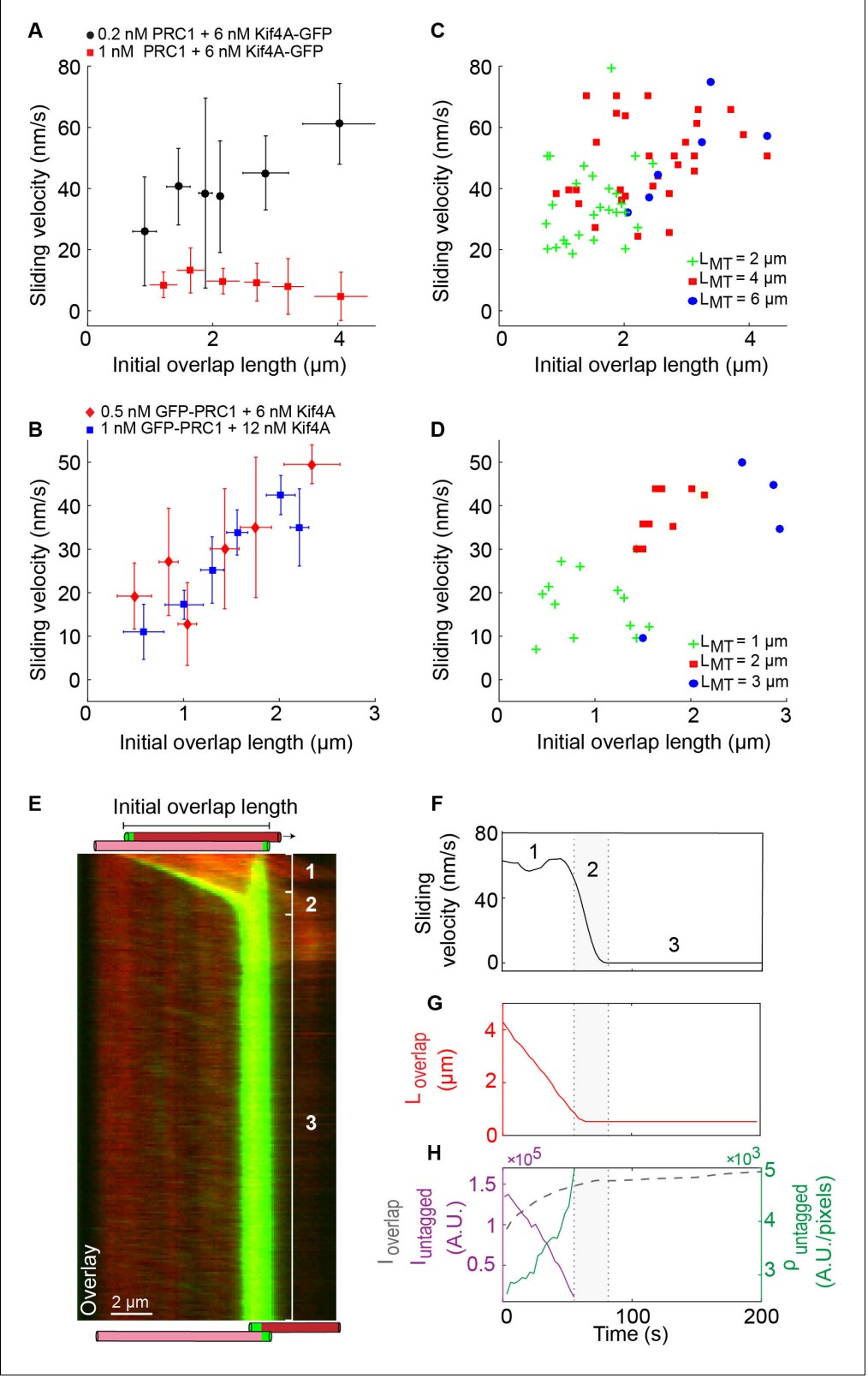

**Figure 4.** Microtubule sliding velocity in the PRC1-Kif4A system scales with initial overlap width. (**A–B**) Binned plots of initial sliding velocity versus the initial overlap length. The initial overlap length between the moving MT and immobilized MT is calculated from the rhodamine MT channel. Sliding velocity is calculated from the constant velocity movement in phase-1. (**A**) Assay conditions: (i) 0.2 nM PRC1 and 6 nM Kif4A-GFP (black; N = 60; Pearson's

*Figure 4 continued on next page*

*Figure 4 continued*

correlation coefficient = 0.54) and (i) 1 nM PRC1 and 6 nM Kif4A-GFP (red; N = 42; Pearson's correlation coefficient = 0.03). (**B**) Assay conditions: (i) 0.5 nM GFP-PRC1 and 6 nM Kif4A (red; N = 25; Pearson's correlation coefficient = 0.69) and (ii) 1 nM GFP-PRC1 and 12 nM Kif4A (blue; N = 20; Pearson's correlation coefficient = 0.74). (**C–D**) Scatter plot of the average sliding velocity versus the initial overlap length color-coded by moving microtubule length, $L_{MT}$. (**C**) Assay condition: 0.2 nM PRC1 and 6 nM Kif4A-GFP (green: $L_{MT}$ = 2 ± 0.5 μM, red: $L_{MT}$ = 4 ± 0.5 μM, blue: $L_{MT}$ = 6 ± 0.5 μM; N = 60). (**D**) Assay condition: 0.5 nM GFP-PRC1 and 6 nM Kif4A (green: $L_{MT}$ = 1 ± 0.5 μM, red: $L_{MT}$ = 2 ± 0.5 μM, blue: $L_{MT}$ = 3 ± 0.5 μM; N = 25). (**E**) Kymograph shows the relative sliding and stalling in a pair of antiparallel microtubules (red) and associated Kif4A-GFP (green). Assay condition: 0.2 nM PRC1 and 6 nM Kif4A-GFP. Scale bar: 2 μm. The schematic illustrates the position and relative orientation of both the immobilized (pink) and moving (red) microtubules and the end-tags (green) at the beginning and end of the time sequence. (**F**) Time record of the instantaneous sliding velocity of the moving microtubule derived from the kymograph in (**E**). The dashed lines demarcate the three phases observed in the sliding velocity profile: (1) constant phase, (2) slow down and (3) stalling. (**G**) Time record of the overlap length (red; $L_{overlap}$) derived from the kymograph in (**E**). (**H**) Time record of the total fluorescence intensity in the antiparallel overlap (dashed gray; $I_{overlap}$), fluorescence intensity in the untagged region of the overlap (solid purple; $I_{untagged}$), and fluorescence density (intensity per unit overlap length) in the untagged region of the overlap (solid green; $\rho_{untagged}$) derived from the kymograph in (**E**).

DOI: https://doi.org/10.7554/eLife.32595.006

The following source data and figure supplements are available for figure 4:

**Source data 1.** This spreadsheet contains the sliding velocity versus the initial overlap length used to generate the binned plots shown in *Figure 4A–B* and the scatter plots shown in *Figure 4C–D*.

DOI: https://doi.org/10.7554/eLife.32595.011

**Figure supplement 1.** Relative microtubule sliding and the formation of stable antiparallel microtubule overlaps by PRC1 and Kif4A.

DOI: https://doi.org/10.7554/eLife.32595.007

**Figure supplement 1—source data 1.** This spreadsheet contains (1) the average sliding velocity versus the sum of end-tag intensity, $I_{ET1}$ + $I_{ET2}$, used to generate the scatter plot in *Figure 4—figure supplement 1A*, (2) the average sliding velocity versus the sum of end-tag length, $L_{ET1}$ + $L_{ET2}$, used to generate the scatter plot in *Figure 4—figure supplement 1B*, (3) the overlap intensity, $I_{overlap}$, versus the overlap length, $L_{overlap}$, used to generate the scatter plot in *Figure 4—figure supplement 1C*, (4) the average sliding velocity versus the untagged overlap length, $L_{untagged}$, used to generate the scatter plots in *Figure 4—figure supplement 1D-E*, (5) the average sliding velocity versus average untagged density ($\rho_{untagged}$) at phase-1 used to generate the scatter plot in *Figure 4—figure supplement 1F* and (6) the average sliding velocity versus end-tag intensity of Kif4A-GFP used to generate the scatter plot and histogram in *Figure 4—figure supplement 1Q*.

DOI: https://doi.org/10.7554/eLife.32595.012

**Figure supplement 2.** Kymographs and the corresponding quantification of microtubule sliding and stopping events.

DOI: https://doi.org/10.7554/eLife.32595.008

**Figure supplement 3.** Analysis of sliding velocity and initial overlap length at equilibrium.

DOI: https://doi.org/10.7554/eLife.32595.009

**Figure supplement 3—source data 2.** This spreadsheet contains the sliding velocity as a function of initial overlap length at equilibrium used to generate the scatter plot.

DOI: https://doi.org/10.7554/eLife.32595.013

**Figure supplement 4.** Protein density at phase 1–2 transition in microtubule overlaps.

DOI: https://doi.org/10.7554/eLife.32595.010

**Figure supplement 4—source data 3.** This spreadsheet contains untagged density $\rho_{untagged}$, in the microtubule overlap at phase 1-2 transition used to generate the histogram in *Figure 4—figure supplement 4A*.

DOI: https://doi.org/10.7554/eLife.32595.014

These data are consistent with a mechanism in which sliding velocity is set by the total number of sliding-competent PRC1-Kif4A molecules in the untagged region of the antiparallel microtubule overlap.

In order to separate the effect of microtubule length versus antiparallel overlap length on the sliding velocity, we re-plotted the data in *Figure 4A–B* based on the length of the moving microtubule. A scatter plot of the sliding velocity as a function of initial overlap length color-coded by the moving-microtubule length shows that longer microtubules typically form longer initial overlaps that

exhibit faster sliding (*Figure 4C–D*). However, the observation that long microtubules that form short overlaps exhibit slower sliding than long microtubules that form long overlaps (for example: 6 μm microtubules with ~3 μm overlap in *Figure 4C*, blue dots), suggests the dominant contribution to the sliding velocity is from the initial overlap length (*Figure 4C–D*). Our findings indicate that the initial length of the antiparallel overlap can tune the microtubule sliding velocity, such that longer overlaps, which have a greater number of PRC1-Kif4A molecules, slide at a faster rate than shorter microtubule overlaps.

## Examining the time-dependent changes during microtubule sliding in the PRC1-Kif4A system

How does the initial antiparallel overlap set the phase-1 sliding velocity? To gain insights into this question, we examined the time-dependent changes in the sliding velocity as a function of overlap length (0.2 nM PRC1 + 6 nM Kif4A-GFP). We focused on a subset of events where the reduction in overlap length ($L_{overlap}$) begins from the initial time point of recording (34/62 events). We find that in all of these events, movement velocity is constant during phase-1 as the overlap length continually shrinks during sliding (*Figure 4E–H* and *Figure 4—figure supplement 1G–P*, *2F-H and N-P*). For example, in the kymograph shown in Figure 4E, the reduction in microtubule overlap length begins at 0 s (*Figure 4G*, solid red line) but a significant reduction in the velocity is not seen until 60 s (*Figure 4F*, solid black line; see also *Figure 4—figure supplement 1N–P*, *2F-H and N-P*).

The observation that microtubule sliding occurs at a constant velocity even as the overlap shrinks, raises the following question: do the number of Kif4A molecules in the overlap change during relative sliding? Analyses of GFP intensity versus time showed that in phase-1, the total amount of Kif4A-GFP in the microtubule overlap ($I_{overlap} = I_{end-tagged} + I_{untagged}$) initially increases and then reaches a constant level that is maintained during all three phases (dashed gray line) (*Figure 4H* and *Figure 4—figure supplement 1N-P* and *2*). We examined events in which phase-1 sliding continued after the establishment of constant motor number and find that sliding velocity scales with initial overlap length under equilibrium conditions (*Figure 4—figure supplement 3*). The observation that $I_{overlap}$ remains constant even as the overlap length reduces suggests that the number of Kif4A molecules per unit length (density) increases in the shrinking overlap during sliding. Is this increase in density in the overlap region entirely due to end-tag formation or is there an increase in the number of Kif4A molecules per unit length of the untagged overlap during microtubule sliding? To answer this question, we quantitatively analyzed the Kif4A-GFP intensity in the untagged region of the overlap during sliding. We find that while the total number of Kif4A-GFP molecules in the untagged overlap region ($I_{untagged}$) (*Figure 4H*; solid purple curve) decreases with shrinking overlap, the Kif4A-GFP density ($\rho_{untagged}$) (*Figure 4H*; solid green curve; fluorescence intensity/pixel) increases.

These data, together with the observation that the number of molecules in the end-tag does not contribute significantly to sliding velocity (*Figure 4—figure supplement 1Q* and *Figure 4—figure supplement 1A–B*), indicate that the velocity of microtubule sliding in the PRC1-Kif4A system is determined by the initial width of the PRC1-crosslinked antiparallel overlap, which sets the total number of sliding competent molecules in the untagged overlap. During microtubule sliding at phase-1, the increase in the density of motor molecules could compensate for the reduction in overlap length and the number of sliding-competent motors, ensuring that microtubule-movement proceeds at a constant velocity.

## Mechanism for transition from constant velocity sliding to slow-down and stalling in the PRC1-Kif4A system

What is the mechanism underlying the shift from constant to decreasing sliding velocity (phase-1 to phase-2) observed in these experiments? First, we considered the possibility that sliding slows down due to the transition from constant to decreasing overlap length and a concomitant increase in protein density as the moving microtubule slides past the immobilized microtubule. Such a mechanism has been described previously for the slowdown of microtubule sliding in the Ase1-Ncd system (*Schizosaccharomyces pombe* Ase1 and *Drosophila* Kinesin-14, Ncd), and by the human kinesin-14 HSET (*Braun et al., 2017*; *Braun et al., 2011*). As discussed in the previous section, our results show that the transition from phase-1 to phase-2 does not coincide with a shift from constant to decreasing overlap length during sliding, and there is no inverse correlation between Kif4A-GFP density and

the sliding velocity during phase-1 movement under the same experimental condition (See *Figure 4E–H*, *Figure 4—figure supplements 1G–P and* and *2*). Next, we considered the possibility that the transition from sliding to slowdown coincides with the end-tags on the moving and immobilized microtubules arriving at close proximity. We hypothesized that at the high-density region proximal to the end-tag, sliding first slows down before the microtubule movement is completely halted. We noticed that in 50% of the events (47/98; 0.2 nM PRC1 + 6 nM Kif4A-GFP), the transition from phase 1 to 2 occurs when the end-tags have nearly merged and our image analysis algorithm does not resolve the two end-tags (examples in *Figure 4E–H* and *Figure 4—figure supplement 1G–P*). We reasoned that if high protein concentration proximal to end-tags is the reason for slow-down, the protein density at the phase-1 to phase-2 transition should be similar under different experimental conditions. Kif4A-GFP density measurement and fluorescence line-scan analysis at the phase-1 to phase-2 transition time-points in two experimental conditions show that this is indeed the case (*Figure 4—figure supplement 4A–B*). Furthermore, comparison of the phase-1 to phase-2 transition-point intensity with the average end-tag intensity suggests that on average slowdown occurs when the intensity is >70% of the average end-tag intensity (*Figure 4—figure supplement 4B–C*). These observations suggest that sliding slows down proximal to end-tags, when the untagged overlap is short and the motors encounter a high-density region, where stepping is inhibited.

## The size of stable antiparallel overlaps established by PRC1 and Kif4A are determined by microtubule length and protein concentration

Our findings suggest that in the PRC1-Kif4A system, a stable antiparallel overlap is formed when the end-tags on both microtubules merge during relative sliding. We hypothesized that if stable overlaps form upon the collision of end-tags on the moving and immobilized microtubules, then the final overlap length ($L_{FO}$) should be determined by the sum of the two end-tag lengths ($L_{ET1} + L_{ET2}$) (*Figure 5A*). Consistent with this, the average ratio of the $\frac{L_{FO}}{L_{ET1} + L_{ET2}}$ at 1 nM PRC1 + 6 nM Kif4A-GFP is ~1 (*Figure 5B*, red). Similar results were observed when the experiments were performed under three different conditions with GFP-PRC1 and untagged Kif4A (*Figure 5—figure supplement 1*). These findings indicate that the width of the stable microtubule overlap established by PRC1 and Kif4A is approximately equal to the sum of the end-tag lengths on both microtubules. At the lowest concentration of PRC1 tested (0.2 nM PRC1 + 6 nM Kif4A-GFP), the final overlap length was shorter than the $L_{ET1} + L_{ET2}$, as indicated by a ratio of < 1 (*Figure 5B*, black). A possible reason is that under these conditions, the end-tagged regions of the microtubules have a greater fraction of unoccupied sites that allows for further sliding and reduction in the overlap length after the collision of end-tags.

Prior work shows that the collective activities of PRC1 and Kif4A on single microtubules result in the formation of end-tags whose size scales with microtubule length (*Subramanian et al., 2013*). This raises the question of whether the width of stable antiparallel overlap established by these proteins depends on the lengths of the two crosslinked microtubules. To examine this, we plotted the final overlap length ($L_{FO}$) as a function of the immobilized microtubule length ($M_{L1}$), moving microtubule length ($M_{L2}$) and sum of both microtubule lengths ($M_{L1} + M_{L2}$). In all three cases, we find that the final overlap length increases linearly with microtubule length (*Figure 5C–E*). The slope of the line is higher at greater PRC1 concentration due to longer end-tags formed under these conditions (*Figure 5E*; 0.2 nM PRC1 + 6 nM Kif4A-GFP, slope = 0.3; 1 nM PRC1 + 6 nM Kif4A-GFP, slope = 0.8). These data suggest that PRC1-Kif4A end-tags act as a barrier to microtubule sliding and establish a stable antiparallel overlap whose size is determined by the microtubule lengths.

## Examination of the mechanisms that ensure stability of the overlaps established by PRC1 and Kif4A

Why does the merging of PRC1-Kif4A end-tags during microtubule sliding result in the formation of a stable antiparallel overlap? It has been shown that the entropic forces induced by Ase1p molecules (*Schizosaccharomyces pombe* PRC1 homolog) can counter the microtubule sliding-associated forces generated by Ncd (*Drosophila* kinesin-14) molecules to establish a stable antiparallel overlap (*Lansky et al., 2015*). Similar observations have been reported with the human kinesin-14 HSET (*Braun et al., 2017*). We therefore examined if entropic forces are generated in the stalled microtubule overlaps established by PRC1 and Kif4A in our experiments. First, we induced the formation of stable overlaps through microtubule sliding and stalling in the presence of GFP-PRC1, Kif4A and

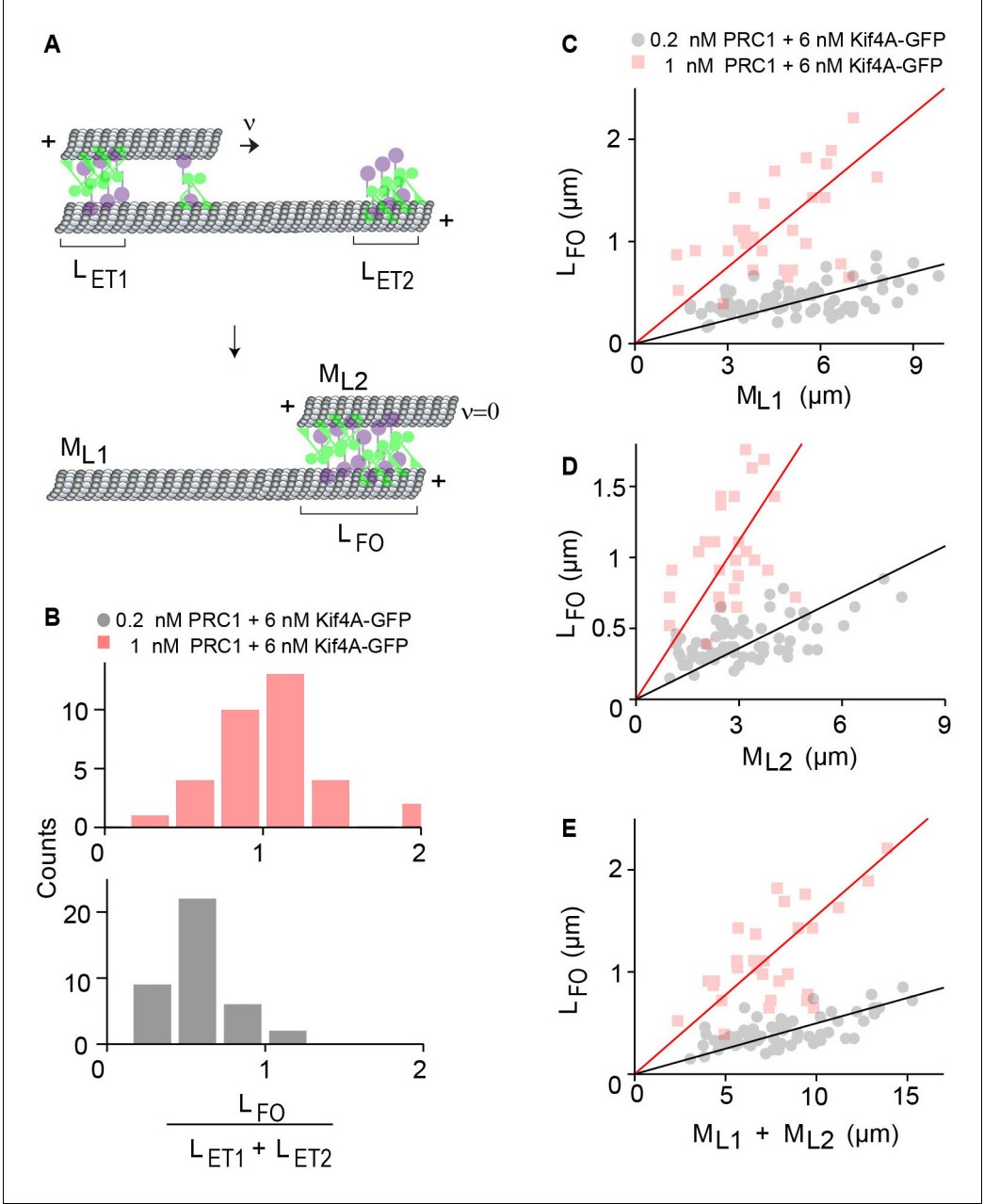

**Figure 5.** The width of the final antiparallel overlap established by PRC1 and Kif4A is determined by end-tag and microtubule lengths. (A) Schematic shows the formation of a stable antiparallel overlap upon collision of the two end-tags and the stalling of relative microtubule sliding. The initial overlap length is the overlap length of the moving MT on the immobilized MT at $t$ = 0. $L_{ET1}$ and $L_{ET2}$ are the lengths of the end-tags consisting Kif4A and PRC1 on the plus-end of each MT. The moving MT with length $M_{L2}$ moves relative to the immobilized MT with length $M_{L1}$, at velocity = $v$. The collision and the stalling of the end-tags form a stable overlap, which is the final overlap length $L_{FO}$ at $v$ = 0. (B) Histograms of the ratio of sum of the end-tag lengths ($L_{ET1}$ + $L_{ET2}$) and final overlap length $L_{FO}$. Assay conditions: (i) 0.2 nM PRC1 and 6 nM Kif4A-GFP (black; N = 39) and (ii) 1 nM PRC1 and 6 nM Kif4A-GFP (red; N = 33). (C–E) Plots of the final overlap length ($L_{FO}$) versus (C) the immobilized microtubule length ($M_{L1}$), (D) moving microtubule length ($M_{L2}$), and (E) and the sum of microtubule lengths ($M_{L1}$ + $M_{L2}$). Assay conditions: (i) 0.2 nM PRC1 and 6 nM Kif4A-GFP (black; N = 68) and (ii) 1 nM PRC1 and 6 nM Kif4A-GFP (red; N = 30). The Pearson's correlation coefficient for (E) is (i) 0.65 and (ii) 0.62.
DOI: https://doi.org/10.7554/eLife.32595.018

The following source data and figure supplements are available for figure 5:

*Figure 5 continued on next page*

*Figure 5 continued*

**Source data 1.** This spreadsheet contains the ratio of the sum of the end-tag lengths ($L_{ET1} + L_{ET2}$) and final overlap length ($L_{FO}$) used to generate the histogram in *Figure 5B* and the final overlap length ($L_{FO}$) versus the immobilized microtubule length ($M_{L1}$), moving microtubule length ($M_{L2}$), and the sum of microtubule lengths ($M_{L1} + M_{L2}$) used to generate the plots in *Figure 5C-E*.

DOI: https://doi.org/10.7554/eLife.32595.020

**Figure supplement 1.** The width of the final antiparallel overlap established by PRC1 and Kif4A is determined by end-tag and microtubule lengths.

DOI: https://doi.org/10.7554/eLife.32595.019

**Figure supplement 1—source data 1.** This spreadsheet contains the ratio of sum of the end-tag lengths ($L_{ET1} + L_{ET2}$) and final overlap length ($L_{FO}$) used to generate the histogram.

DOI: https://doi.org/10.7554/eLife.32595.021

ATP. Next, we washed the assay chamber twice with buffer containing no ATP to remove any unbound protein and nucleotide. Under this 'no-nucleotide' condition, we expect that the PRC1-Kif4A complexes in the microtubule overlap would essentially function as passive crosslinkers. Dual-wavelength time-lapse images were acquired for 10 min immediately following buffer exchange. Image analysis revealed that while PRC1 was retained in the region of the microtubule overlap under these conditions, the width of the antiparallel overlap did not change during the course of the experiment (*Figure 6—figure supplement 1A–B*). The lack of overlap expansion in the PRC1-Kif4A system may be due to the tight binding of the kinesin motor domain to microtubules in the absence of a nucleotide. To address this, we performed the experiment as discussed above, except the final buffer was supplemented with 2 mM ADP, a nucleotide that lowers the kinesin-microtubule affinity. As shown in *Figure 6—figure supplement 1C–F*, no overlap expansion was observed under these conditions. The inclusion of 1 nM PRC1 in addition to 2 mM ADP in the final buffer also did not promote overlap expansion (*Figure 6—figure supplement 1G–H*). Therefore, neither motor deactivation with ADP nor increasing the number of PRC1 molecules is sufficient to induce entropic expansions of measurable magnitude in this system, suggesting that an alternative mechanism is likely responsible for countering the Kif4A-mediated sliding forces in the antiparallel overlap.

We have previously shown that PRC1-Kif4A end-tags on single microtubules hinder motor-protein stepping (*Subramanian et al., 2013*). Therefore, we considered if the collision of end-tags on sliding microtubules generated a stable antiparallel overlap simply by providing a steric block to sliding. To test this hypothesis, we generated stable antiparallel overlaps with PRC1, Kif4A-GFP and ATP, and subsequently exchanged the nucleotide to ADP by buffer exchange (*Figure 6A–C*). As expected, no change in the overlap length was observed upon nucleotide exchange from ATP to ADP. We reasoned that under these experimental conditions, the gradual dissociation of proteins at a slow rate from the overlap should liberate a small fraction of kinesin and PRC1 binding sites on the microtubule (note: intensity analysis suggests a maximum 10% reduction of Kif4A-GFP in 2 min). Therefore, if the moving microtubule had initially stalled due to protofilament crowding, then re-introducing ATP should allow motor-protein stepping and reinitiate microtubule sliding. To test this experimentally, we introduced buffer containing 1 mM ATP (no additional protein) into the chamber 15 min after the ADP exchange step (*Figure 6D*). We find that relative microtubule sliding is reinitiated under these conditions. Analysis of the GFP fluorescence-intensity profile at different time-points post buffer exchange revealed that new end-tags are established during microtubule sliding, which subsequently collide to establish a new stable antiparallel overlap of shorter width (*Figure 6E*).

Together, these findings are consistent with a mechanism in which PRC1-Kif4A end-tags establish stable overlaps by sterically hindering the relative sliding of antiparallel microtubules. Such a 'molecular road-block' based mechanism also provides a simple explanation for the observed correlation between the sum of end-tag lengths and the final overlap length in this system.

## PRC1 and Kif4A align the overlap region between multiple antiparallel microtubules

How do microtubule sliding and stalling by PRC1 and Kif4A shape larger microtubule arrays? To gain insights into this question, we carefully examined the few events (N < 10) where we could clearly observe two microtubules slide relative to a single immobilized microtubule. In these events

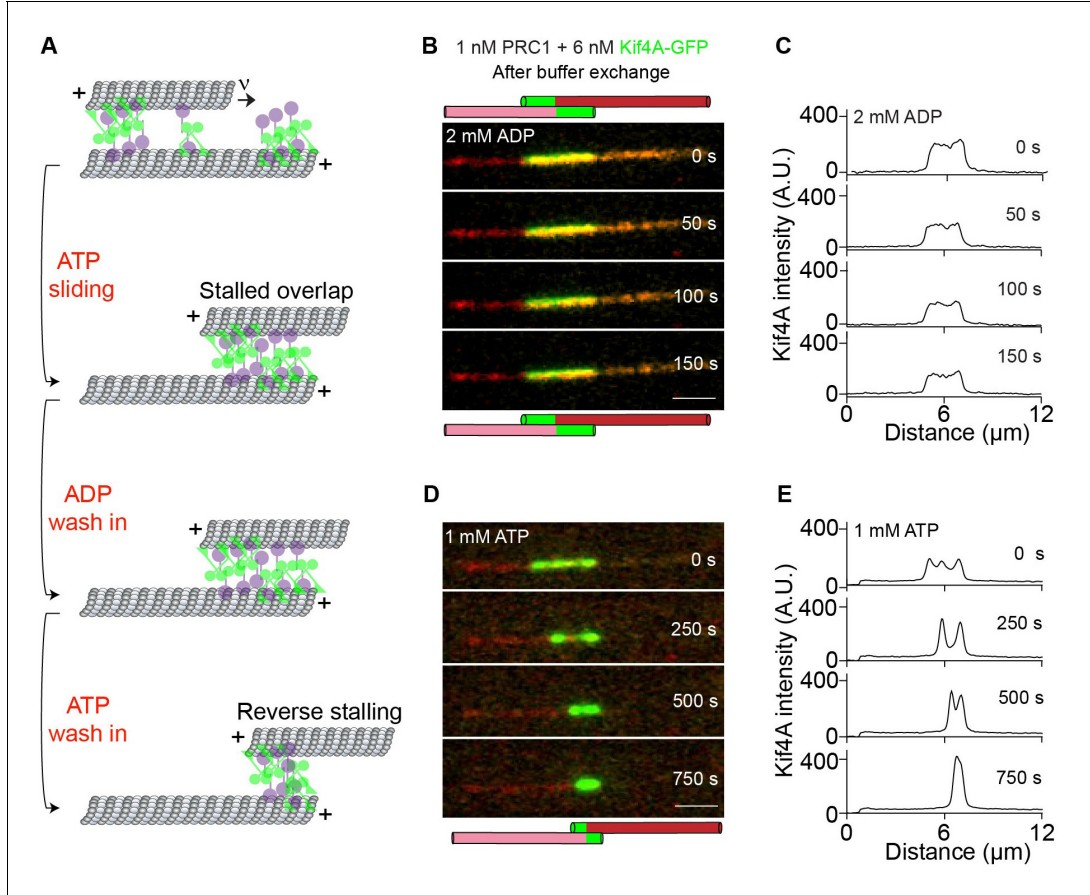

**Figure 6.** Examination of the mechanisms that ensure stability of the overlaps established by PRC1 and Kif4A. (A) Schematic of the ADP and ATP wash-in experiments performed with stalled microtubule overlaps *Figure 6B-E*. (B–E) The following figures are representative dual-channel fluorescence micrographs showing microtubules (red) and associated Kif4A-GFP (green) under different experimental conditions. (B–C) Time-lapse images (B) and corresponding line-scan profiles (C) of Kif4A-GFP fluorescence of a microtubule pair established as in (*Figure 1A* ) and subsequent exchange into a buffer containing 2 mM ADP. (D–E) Time-lapse images (D) and corresponding line-scan profiles (E) of Kif4A-GFP fluorescence of the microtubule pair in (D) after flowing in 1 mM ATP into the chamber. Scale bar: 2 μm.

DOI: https://doi.org/10.7554/eLife.32595.022

The following figure supplement is available for figure 6:

**Figure supplement 1.** Examination of the mechanisms that maintain the stable antiparallel overlaps established by PRC1 and Kif4A.

DOI: https://doi.org/10.7554/eLife.32595.023

(*Figure 7A–C*; 0.2 nM PRC1 + 6 nM Kif4A-GFP), we observed that both the sliding microtubules stall proximal to the plus end-tag on the immobilized microtubule. Another example of such an event in experiments with GFP-labeled PRC1 and unlabeled Kif4A is shown in *Figure 7D–F* (1 nM GFP-PRC1 + 6 nM Kif4A). The data suggest that the formation of end-tags on single microtubules can establish an antiparallel array composed of multiple microtubules with closely aligned plus-ends.

We analyzed five reorganization events where we could reliably measure microtubule and overlap lengths to determine if longer microtubules result in larger final overlaps in these more complex bundles. While we cannot assess the three-dimensional arrangement of the microtubules in the bundles, a simple analysis of microtubule and overlap lengths suggests that in general bundles with longer microtubules are likely to yield longer final overlaps (*Figure 7—figure supplement 1*).

Together, these observations suggest that microtubule sliding and stalling by PRC1 and Kif4A can align multiple antiparallel filaments such that the region of overlap is restricted to the plus-ends of all the microtubules.

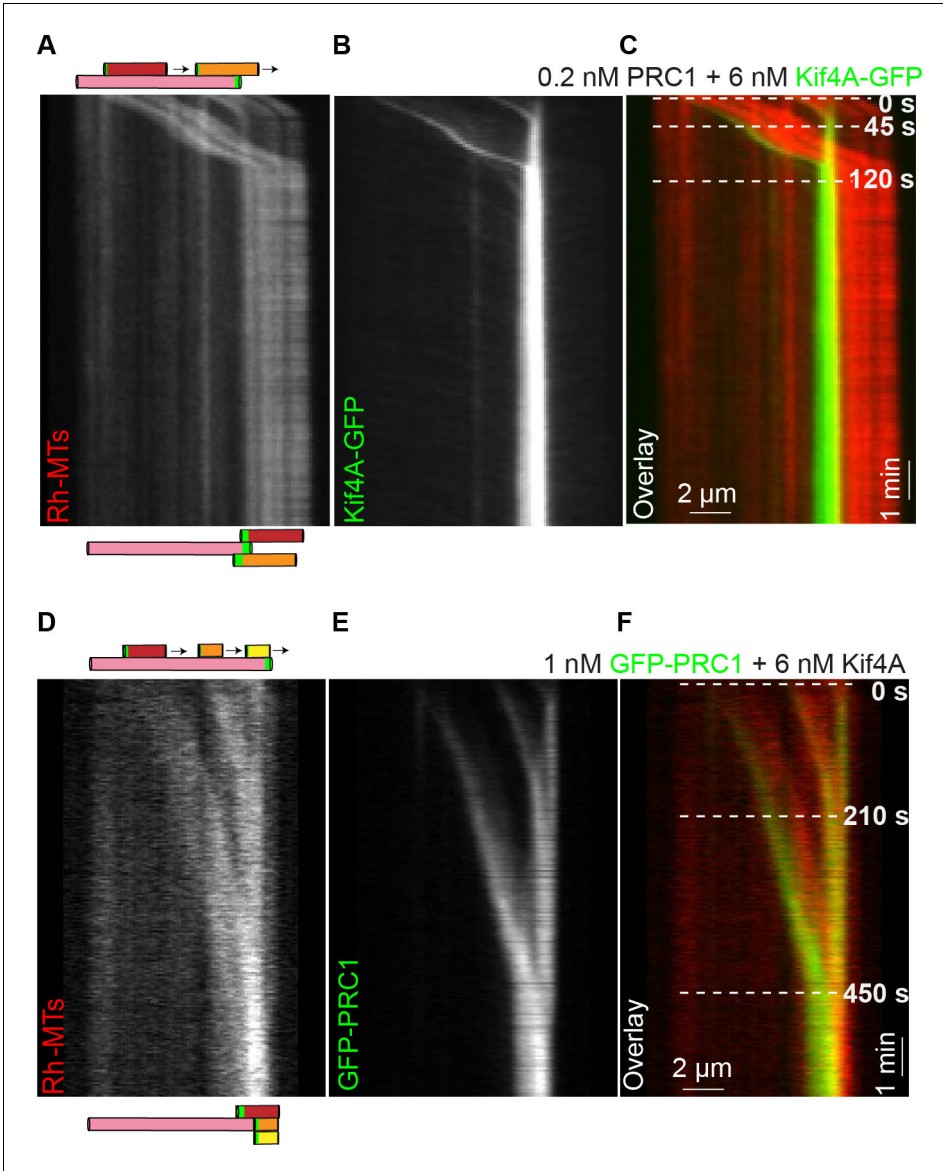

**Figure 7.** Antiparallel array composed of multiple microtubules are aligned at microtubule plus-ends formed by PRC1 and Kif4A. (**A–C**) Kymographs show the relative sliding of two microtubules relative to an immobilized microtubule (**A**), associated Kif4A-GFP (**B**) and the overlay image (red, microtubules; green, Kif4A-GFP) (**C**). Both moving microtubules stall at the plus-end of the immobilized microtubule. Assay condition: 0.2 nM PRC1 and 6 nM Kif4A-GFP. Scale bar: *x*: 2 μm and *y*: 1 min. (**D–F**) Kymographs show the relative sliding of three microtubules relative to an immobilized microtubule (**D**), associated GFP-PRC1 (**E**) and the overlay image (red, microtubules; green, GFP-PRC1) (**F**). All three moving microtubules stall at the plus-end of the immobilized microtubule. Assay condition: 1 nM GFP-PRC1 and 6 nM Kif4A. Scale bar: *x*: 2 μm and *y*: 1 min.

DOI: https://doi.org/10.7554/eLife.32595.024

The following source data and figure supplements are available for figure 7:

**Figure supplement 1.** Multiple microtubule sliding by PRC1 and Kif4A.

DOI: https://doi.org/10.7554/eLife.32595.025

**Figure supplement 1—source data 1.** This spreadsheet contains the final overlap length ($L_{FO}$) versus the sum of microtubule lengths ($M_{L1} + M_{L2}$) used to generate the plot.

DOI: https://doi.org/10.7554/eLife.32595.026

## Discussion

Pairs of crosslinked antiparallel microtubules are fundamental structural units in diverse microtubule-based architectures (*Dogterom and Surrey, 2013*; *Subramanian and Kapoor, 2012*). Our findings provide insights into how the geometrical features of antiparallel microtubule arrays can be 'decoded' by PRC1-Kif4A complexes to govern the dynamics, stability and architecture of microtubule networks.

On the basis of our observations, we propose a mechanism for the organization of stable microtubule length-dependent antiparallel overlaps by the collective activities of PRC1 and Kif4A. PRC1 specifically crosslinks and preferentially localizes to the region of overlap between two antiparallel microtubules (*Figure 8A*) (*Bieling et al., 2010*; *Subramanian et al., 2010*). Kif4A is recruited to the antiparallel overlap through direct interaction with PRC1 (*Bieling et al., 2010*; *Subramanian et al., 2013*). The highly processive movement of PRC1-Kif4A complexes on microtubules and the slow dissociation of these proteins from microtubule plus-ends result in the formation of 'end-tags', which are highly crowded regions in which motor stepping is inhibited (*Figure 8A*) (*Subramanian et al., 2013*; *Leduc et al., 2012*). In addition, the activity of PRC1-Kif4A complexes within antiparallel overlaps results in robust relative microtubule sliding (*Figure 8A*). During sliding, as the moving microtubule moves past the length of the immobilized microtubule, the distance between the end-tags at the plus-ends of both microtubules begins to shrink (*Figure 8A*). Microtubule movement first slows down and then stalls when the two end-tags arrive at close proximity during relative sliding, resulting in the formation of a stable antiparallel overlap (*Figure 8A*).

## Organization of stable antiparallel overlaps by PRC1 and Kif4A

Non-motor crosslinking proteins are primarily thought to contribute to the size and stability of microtubule arrays by opposing the active forces generated by motor proteins (*Peterman and Scholey, 2009*). Such a mechanism has been proposed for the formation of stable overlaps by the collective activity of the *Drosophila* Kinesin-14, Ncd and the *Schizosaccharomyces pombe* Ase1 (*Braun et al., 2011*). Similarly, the human kinesin-14 HSET, can both generate active and counter-acting entropic forces to generate stable antiparallel overlaps (*Braun et al., 2017*). In contrast, we propose that the predominant mechanism that leads to the formation of a stable overlap in the PRC1-Kif4A system is steric hindrance to motor stepping at regions of high protein-density for the following reasons. First, sliding microtubules come to a stall when the PRC1-Kif4A end-tags arrive at close proximity. Steric hindrance at an end-tag is expected to be high because the PRC1 and Kif4A binding sites at this region of the microtubule are likely to be nearly saturated (*Subramanian et al., 2013*; *Leduc et al., 2012*). Further hindrance to stepping at the crowded end-tags can arise due to the partial overlap of the tubulin-binding interfaces of PRC1 and kinesin (*Kellogg et al., 2016*). Second, the accumulation of Kif4A alone at microtubule ends can stop sliding, and full-length Kif4A does not have a C-terminus non-motor domain that binds tightly to microtubules and is therefore unlikely to generate significant frictional forces (*Figure 2C–D*). Third, we do not observe entropic force driven expansion of stable overlaps formed by PRC1 and Kif4A when the motor is deactivated (*Figure 6*). Fourth, we do not observe an inverse correlation between sliding velocity and changes in protein density during phase-1 sliding as is predicted in models where the buildup of opposing forces in shrinking overlaps stall microtubule sliding (*Figure 4—figure supplement 1F*). Finally, previous studies have shown that the frictional forces generated by PRC1 are low, and the speed of Kif4A movement on single microtubules is not affected by PRC1 (*Duellberg et al., 2013*; *Forth et al., 2014*). Combined with the observation that the number of PRC1 molecules is less than Kif4A motors at the microtubule overlap in our experiments (*Figure 8—figure supplement 1*), it is unlikely that stable overlaps can be formed solely through the opposition of Kif4A-generated forces by PRC1 at end-tags. For similar reasons as stated above, steric hindrance is likely to play an important role in switching from sliding at constant velocity to slowdown at the phase-1 to phase-2 transition (*Figures 3C* and *4F*), and the reduction in average sliding velocity at high PRC1 concentrations (*Figure 3D* and *Figure 3—figure supplement 1*). In the future, it will be interesting to determine the contribution of frictional forces in the slowdown of sliding in the PRC1-Kif4A system. Overall, in contrast to mechanisms where stable microtubule arrays are organized through opposing forces generated by a pair of motor and non-motor protein, the PRC1-Kif4A system reveals an alternative mechanism in which a motor and a

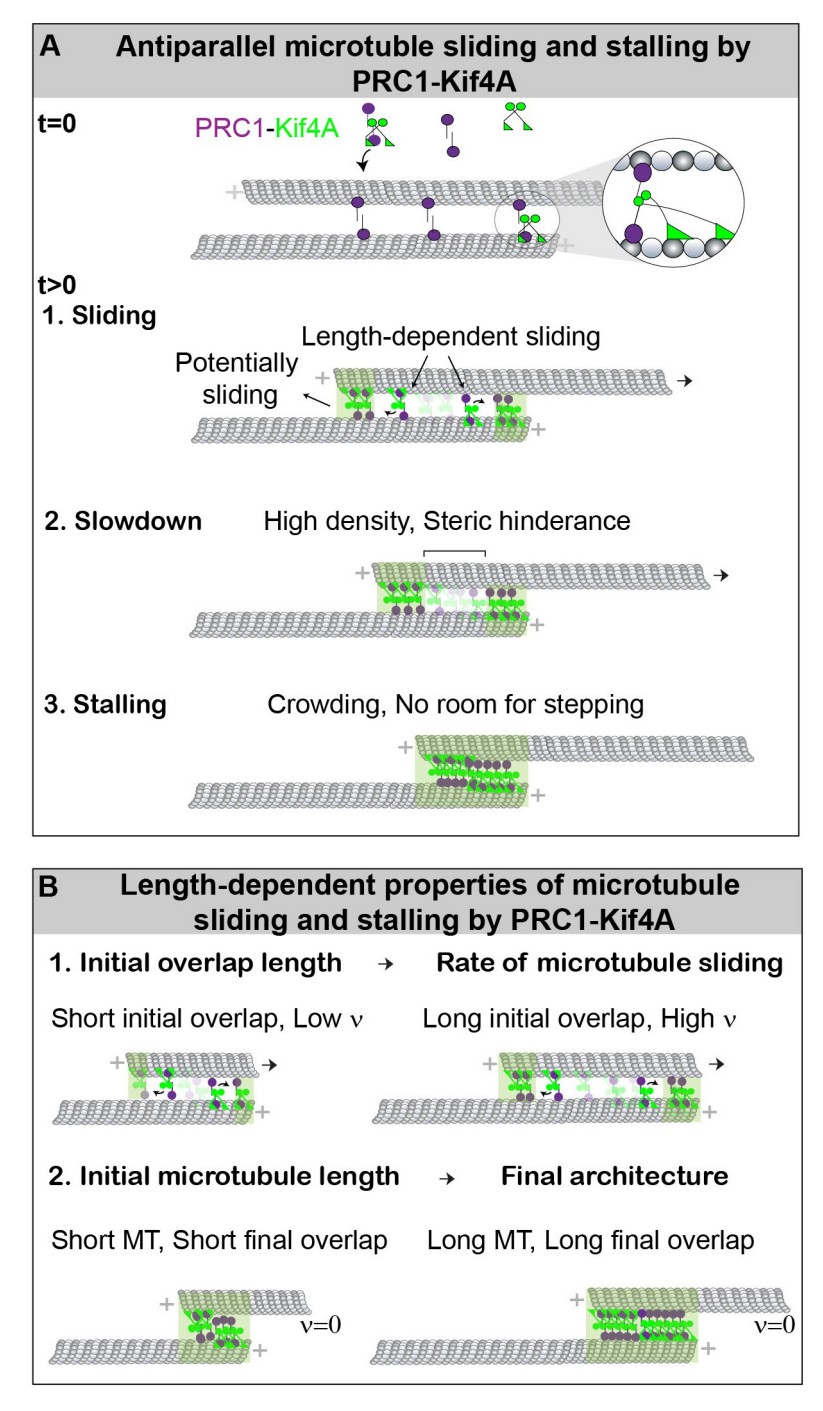

**Figure 8.** Model for the length-dependent sliding by the collective activity of PRC1 and Kif4A. (**A**) Mechanism of microtubule sliding and stalling by PRC1 and Kif4A. At the initial state, $t = 0$, the 'immobilized' and 'moving' microtubules are crosslinked by PRC1 to form an antiparallel overlap. The zoomed-in view shows the proposed molecular configuration of the cross-bridging PRC1-Kif4A complex in a microtubule overlap. At $t > 0$, Kif4A molecules are introduced into the solution, which form a complex with PRC1. This initiates the formation of PRC1-Kif4A end-tags at the plus-ends of both microtubules as well as relative sliding of the moving microtubule. The sliding of microtubules is most likely due to the cross-bridging molecules in the untagged overlap. The slowdown is likely due to the high density of molecules and steric hindrance to motor stepping when the end-tags arrive at close proximity. This eventually halts movement when the end-tags merge, and a stable overlap is established. (**B**) The schematic shows the length-dependent properties of initial microtubule sliding and subsequent stalling of

*Figure 8 continued on next page*

*Figure 8 continued*

overlapping antiparallel microtubules established by PRC1 and Kif4A. Our experiments show that: (1) Microtubules that form shorter initial overlaps slide with lower velocity than microtubule pairs that form longer initial overlaps. (2) Since the size of PRC-Kif4A end-tags scale with microtubule length, shorter microtubules form a short overlap and the longer microtubules form a long overlap.

DOI: https://doi.org/10.7554/eLife.32595.027

The following source data and figure supplements are available for figure 8:

**Source data 1.** This table contains the sliding efficiencies calculated using *Equation 3* for $a$ = 1-10%.

DOI: https://doi.org/10.7554/eLife.32595.030

**Figure supplement 1.** Average untagged protein density in microtubule overlaps.

DOI: https://doi.org/10.7554/eLife.32595.028

**Figure supplement 1—source data 1.** This spreadsheet contains the average untagged density ($\rho_{untagged}$) used to generate the histogram.

DOI: https://doi.org/10.7554/eLife.32595.031

**Figure supplement 2.** Estimation of the sliding efficiency of PRC1-Kif4A molecules.

DOI: https://doi.org/10.7554/eLife.32595.029

**Figure supplement 2—source data 2.** This spreadsheet contains the microtubule sliding velocity versus the initial microtubule overlap length data used to generate the scatter plot in *Figure 8—figure supplement 2A* and the estimation of the number of molecules/µm used to generate the histogram in *Figure 8—figure supplement 2B*.

DOI: https://doi.org/10.7554/eLife.32595.032

non-motor protein act synergistically on antiparallel microtubules to first promote relative sliding and then stall microtubule movement by forming a molecular roadblock.

Some of the distinct features of the roadblock mechanism are as follows. First, in this system, stable arrays can be established under conditions where the non-motor:motor protein ratio may not be not sufficiently high to achieve force-balance. This is particularly advantageous in the case of interacting proteins, such as PRC1 and Kif4A, where increasing the concentration of PRC1 leads to a concomitant increase in both the levels of motor and non-motor proteins further shifting the force-balance point. Second, this system allows for robust relative sliding until the end-tags collide. This in turn leads to the establishment of stable antiparallel overlaps that are spatially restricted to microtubule plus-ends. Third, it provides a simple mechanism by which the formation of length-dependent PRC1-Kif4A end-tags on single microtubules can be readily translated to the organization of microtubule overlaps whose size scales with microtubule length (*Figure 8B*).

## Mechanism of overlap-length dependent sliding by PRC1 and Kif4A

Thus far, the best-studied systems of microtubule sliding are those mediated by crosslinking tetrameric kinesins such as Eg5 or dimeric motors that have non-motor microtubule binding domains such as kinesin-14 (*Fink et al., 2009*; *Kapitein et al., 2005*; *Korneev et al., 2007*; *Valentine et al., 2006*; *Braun et al., 2009*). We find that (i) Kif4A alone does not promote antiparallel microtubule crosslinking and (ii) the C-terminus tail of Kif4A does not interact with microtubules with binding affinities comparable to other microtubule binding domains (*Figure 2A*). In addition, currently there is no evidence for significant oligomerization of full-length Kif4A in the micromolar concentration range (~10 µM). Based on these observations, we consider two alternative modes of microtubule crosslinking and sliding by Kif4A. First, we find that in experiments with Kif4A alone, end-tags on one microtubule can contact and move along another microtubule (*Figure 2C and D*). We speculate that since end-tags are highly crowded regions, a fraction of Kif4A molecules, especially those at the very tip of the microtubule, may adopt a one-head bound state (as is seen in EM experiments performed at high concentrations of kinesin dimers (*Hoenger et al., 2000*; *Hirose et al., 2000*), and collectively mediate microtubule sliding. This is similar to a recent proposal for dynein driven relative-microtubule sliding to form minus-end asters (*Tan et al., 2018*). Alternatively, it is possible that the high concentration of Kif4A at end-tags may promote intermolecular interactions between motors or binding between the Kif4A C-terminus and microtubule despite these being low-affinity interactions. However, this class of mechanisms, where Kif4A alone is responsible for crosslinking and sliding, is less likely in the untagged overlaps that have lower PRC1 and Kif4A densities. Instead, the C-terminus of Kif4A is likely to interact with the N-terminus of PRC1 ($K_d$ = 0.3 µM)

(*Subramanian et al., 2013*), and the PRC1-tethered Kif4A complexes within the untagged region of the antiparallel overlap drive sliding. However, PRC1 and Kif4A are unlikely to form a stable long-lived complex. Consistent with this, we have not been able to isolate stable PRC1-Kif4A complexes by gel filtration or visualize complexes by single-particle EM (unpublished). However, PRC1-Kif4A complexes have a longer microtubule lifetime than either protein alone, suggesting that PRC1 and Kif4A molecules likely undergo dynamic dissociation and re-association on microtubules (*Bieling et al., 2010*; *Subramanian et al., 2013*). As we discuss below, cross-bridging and sliding by the dynamic complex is likely to contribute to the observed length-dependent sliding in this system.

Surprisingly, we find that the sliding velocity in the PRC1-Kif4A system is proportional to the initial antiparallel microtubule overlap length (*Figure 8B*). While the scaling of movement velocity with motor number has been proposed for microtubule-based transport of cargoes in the cellular cytoplasm (*Hill et al., 2004*; *Kural et al., 2005*; *Levi et al., 2006*), it is not typically observed in microtubule sliding by an ensemble of processive kinesins in in vitro assays. One proposed reason is the low viscous drag experienced by the moving microtubule in aqueous buffers relative to the intracellular environment, and the high magnitude of forces generated by kinesin molecules (*Braun et al., 2009*; *Gagliano et al., 2010*). However, this study and in vitro reconstitution experiments with mixtures of Ncd and Ase1 or with the kinesin-14 HSET alone show that scaling of sliding velocity with microtubule overlap length can be observed in the absence of substantial external viscous drag forces (*Braun et al., 2017*; *Braun et al., 2011*). However, the mechanisms by which length-dependent sliding is achieved in these systems differ. In both the Ncd-Ase1 and HSET systems, reduction in sliding velocity within shrinking overlap occurs concurrently with an increase in protein density (*Braun et al., 2017*; *Braun et al., 2011*). In these systems, increasing Ase1 density opposes Ncd-generated forces, and increasing HSET density generates an entropic force that counteracts its movement. However, in the PRC1-Kif4A system, the initial velocity is constant and does not change with reducing overlap length or changes in motor density during sliding (examples in *Figure 4E–H*, *Figure 4—figure supplements 1G–P* and *2*). In this constant velocity phase, the rate of microtubule movement is determined by the initial overlap width, which determines the number of associated motor molecules. Therefore, the mechanism of overlap length-dependent sliding by PRC1 and Kif4A is different from systems that involve an increase in forces that oppose motor movement in shrinking overlaps during microtubule sliding.

How might the microtubule sliding velocity scale with initial overlap length? Hints to a possible mechanism come from recent studies that investigate how the physical properties of cargoes impact microtubule sliding by motor proteins (*Conway et al., 2012*; *Grover et al., 2016*). For example, when kinesin motors are anchored to a diffusive lipid surface instead of a rigid glass coverslip, the gliding velocity of attached microtubules is dependent on the number of motor molecules (*Grover et al., 2016*). We consider the nature of the 'cargo' borne by the Kif4A molecule. In the untagged overlap of the PRC1-Kif4A system, the cargo is a microtubule polymer that is connected to the Kif4A motor via a PRC1-mediated linkage. The measured dissociation constant between the C-terminus of Kif4A and the N-terminus of PRC1 is 0.3 μM, suggesting that these proteins do not form a stable long-lived complex. Therefore the 'moving' microtubule is likely to be loosely coupled to the kinesin walking on an immobilized microtubule, and each 8 nm step of the kinesin does not translate into an 8 nm displacement of the moving microtubule. Other factors, such as force-dependent dissociation of PRC1-microtubule interaction during kinesin stepping, will further reduce the coupling between the two microtubules, and result in the reduction in the 'sliding efficiency', and consequently, the sliding velocity in the PRC1-Kif4A system.

We can define the sliding efficiency (*S*), as the fraction of time spent by a PRC1-Kif4A complex bound to both microtubules. In this sliding-competent conformation, PRC1 is bound to Kif4A and the moving microtubule, and Kif4A is bound to the immobilized microtubule. This formulation is analogous to the 'duty ratio' of a motor, $f$, which is the fraction of the catalytic cycle that a motor head spends attached to a microtubule and undergoes a working stroke (*Hancock and Howard, 1998*; *Howard, 1997*). However, in the PRC1-Kif4A system, reduction in sliding efficiency due to the uncoupling of motor stepping from microtubule sliding can occur either: (i) during the kinesin ATPase cycle or (ii) between two catalytic cycles. In order to estimate the sliding efficiency for microtubule sliding by PRC1-Kif4A, we followed the method in Ref. (*Hamasaki et al., 1995*), and first fitted the microtubule sliding velocity as a function of initial microtubule overlap length (0.2 nM PRC1 + 6 nM Kif4A-GFP) to the following equation,

$$v/v_0 = L/(L + K_L) \tag{1}$$

where $v$ is the microtubule sliding velocity, $v_o$ is the maximal velocity, $L$ is the microtubule overlap length and $K_L$ is the microtubule overlap length when $v = 0.5v_0$. Here, $v$ and $L$ were obtained from experimental data and $v_o$ and $K_L$ were obtained from the fit (*Figure 8—figure supplement 2A*).

Next, using $v_o$ and $K_L$ from above, the sliding efficiency for PRC1-Kif4A, ($S$) was estimated in a manner similar to calculation of motor duty ratios (*Uyeda et al., 1990*). $S$ represents the probability that a PRC1-Kif4A complex is in a crosslinking conformation compatible with sliding. For simplicity, here we assume that the formation of this crosslinked conformation is sufficient for sliding and do not consider other features of the system such as steric hindrance to motor stepping. Consequently, the probability that the filament is propelled by at least one PRC1-Kif4A complex is $\{1 - (1 - S)^N\}$, where $N$ is the number of available sliding competent complexes (*Uyeda et al., 1990*). Therefore:

$$v = v_0\left(1 - (1-S)^N\right) \tag{2}$$

Since $K_L$ is defined as the microtubule overlap length where $v = 0.5v_0$, *Equation 2* leads to

$$0.5 = \left(1 - (1-S)^{N(K_L)}\right) \tag{3}$$

$N$ can be estimated as

$$N(K_L) = \frac{aK_L}{\delta} \tag{4}$$

where $a$ is the experimentally estimated fractional occupancy of sliding-competent motors (molecules per unit length) and $\delta = 8$ nm is the length of a single binding site on microtubules. Based on the experimental fluorescence intensity measurements (see Materials and methods and *Figure 8—figure supplement 2B*), the occupancy of sliding competent PRC1-Kif4A molecules ($a$) is estimated to be 1% if PRC1 molecules can crosslink to all 13 microtubule protofilaments of the moving microtubule and 10% assuming that effective crosslinks are only formed with one protofilament. Considering the molecular structure of PRC1, the values are likely to be in the 1-10% range. From the values of $N$ obtained for $a$ = 1-10%, sliding efficiencies, $S$, calculated using *Equation 3* were between 0.3-0.03 (See *Figure 8—source data 1*). These calculations provide an approximate estimation of the sliding efficiency because (i) we do not reach saturation sliding velocity in our experiments and (ii) the number of sliding-competent molecules is obtained from fluorescence intensity measurements. As a comparison, the estimated values of $S$ are on the same order of magnitude as the duty ratio of motors that exhibits filament length-dependent movement velocities, such as the *Paramecium* 22S dynein ($f$ = 0.01) (*Hamasaki et al., 1995*), β dynein ($f$ = 0.0050) (*Imafuku et al., 1997*), myosin ($f$ = 0.050) (*Uyeda et al., 1990*), and NcKin3 ($f$ = 0.03) (*Adio et al., 2006*). Together, our findings suggest that microtubule sliding by the PRC1-Kif4A complex can be considered as microtubule movement driven by an ensemble of low sliding efficiency motors, which results in the scaling of microtubule sliding velocity with antiparallel overlap length.

## Concluding remarks

A defining architectural feature of the spindle midzone is a stable antiparallel microtubule array with overlapping plus-ends (*Bratman and Chang, 2008*; *Glotzer, 2009*). While we currently do not know if sliding of end-tagged microtubules by PRC1 and Kif4A contributes to the midzone organization, several properties of this system are relevant to the organization of cell biological structures both during mitosis in eukaryotes and in interphase cells of yeast and plant cells (*Dogterom and Surrey, 2013*; *Subramanian and Kapoor, 2012*). The relative sliding of PRC1-crosslinked microtubules by motor proteins such as Cin8 and Kip3p in budding yeast and KLP61F in Drosophila is thought to mediate the spindle elongation during early anaphase and contribute to defining the overlap width (*Sharp et al., 1999*; *Straight et al., 1998*; *Pellman et al., 1995*; *Su et al., 2013*). Antiparallel sliding of PRC1-crosslinked inter-kinetochore bridges is required for proper chromosome segregation (*Polak et al., 2017*; *Vukušić et al., 2017*). At least one of the motors involved in this process is centralspindlin, which also interacts with PRC1. Initial overlap length-dependent sliding could be advantageous in ensuring that microtubules of different lengths arrive at similar rates to the plus-ends of

the template microtubule within arrays. The accumulation of proteins at microtubule plus-ends, including multiple mitotic kinesins, is thought to effectively concentrate cytokinesis factors proximal to the site of cell cleavage (*Shrestha et al., 2012*; *Glotzer, 2009*; *Canman et al., 2003*). Microtubule-end localization of PRC1 has been reported during monopolar cytokinesis (*Subramanian et al., 2013*; *Shrestha et al., 2012*), and in bipolar cells (*Shrestha et al., 2012*; *Shannon et al., 2005*), and both PRC1 and Kif4A have been described to co-localize at the ends of astral arrays of *Xenopus* egg extracts (*Nguyen et al., 2018*). These dense regions of proteins at microtubule ends may contribute to the alignment and stability of antiparallel microtubule arrays. Furthermore, the roadblock mechanism may be utilized by other kinesins that accumulate at microtubule plus-ends for the organization of stable arrays of defined size and geometry (*Shimamoto et al., 2015*; *Braun et al., 2017*; *Bieling et al., 2010*; *Braun et al., 2011*). Our biophysical analyses suggest that the geometrical features of the microtubules in the spindle can in turn tune the activity of associated proteins and regulate the geometry and stability of cellular microtubule arrays such as the spindle midzone.

In summary, our studies show how two microtubule-associated proteins, each with its own distinct filament binding properties, can act collectively to 'measure' the geometrical features of microtubules arrays and 'translate' them to generate well-defined mechanical and structural outputs. Filament crosslinking, relative-sliding and molecular crowding are likely to represent general features of a number of biological polymers, such as actin filaments and nucleic acids, that are dynamically organized during different cellular processes. The mechanism revealed here can therefore represent general principles that regulate the size and dynamics of cellular architectures built from different polymers.

## Materials and methods

### Protein purification
Recombinant proteins used in this study (PRC1, PRC1-GFP, Kif4A and Kif4A-GFP, Kif4A(C-term; aa: 733–1232) were expressed and purified as described previously (*Subramanian et al., 2013*; *Subramanian et al., 2010*).

### Microtubule polymerization
GMPCPP polymerized and taxol-stabilized rhodamine-labeled microtubules were prepared with and without biotin tubulin as described previously (*Subramanian et al., 2013*; *Subramanian et al., 2010*). Briefly, GMPCPP seeds were prepared from a mixture of unlabeled bovine tubulin, X-rhodamine-tubulin and biotin tubulin, which were diluted in BRB80 buffer (80 mM PIPES pH 6.8, 1.5 mM MgCl$_2$, 0.5 mM EGTA, pH 6.8) and mixed together by tapping gently. The tube was transferred to a 37°C heating block and covered with foil to reduce light exposure. Non-biotinylated microtubules and biotinylated microtubules were incubated for 20 min and 1 hr 45 min, respectively. Afterwards, 100 µL of warm BRB80 buffer was added to the microtubules and spun at 75000 rpm, 10 min, and 30°C to remove free unpolymerized tubulin. Following the centrifugation step, the supernatant was discarded and the pellet was washed by round of centrifugation with 100 µL BRB80 supplemented with 20 µM taxol. The pellet was resuspended in 16 µL of BRB80 containing 20 µM taxol and stored at room temperature covered in foil.

### Microtubule Co-sedimentation assay
Kif4A (C-term; 0.8 µM) was incubated with microtubules (0–8 µM) for 15 min at room temperature in motility buffer supplemented with 0.1 µg/µL BSA and then subject to sedimentation in TLA 120.1 rotor (Beckman Coulter) at 90000 rpm for 15 min at 27°C. After re-suspending the pellet, the amount of protein in pellet and supernatant was analyzed by SDS PAGE and the Coomassie-stained bands were quantified (LI-COR Odyssey). Full-length PRC1 was included as a control.

### In vitro fluorescence microscopy assay
The microscope slides (Gold Seal Cover Glass, 24 × 60 mm, thickness No.1.5) and coverslips (Gold Seal Cover Glass, 18 × 18 mm, thickness No.1.5) were cleaned and functionalized with biotinylated PEG and non-biotinylated PEG, respectively, to prevent nonspecific surface sticking, according to standard protocols (*Subramanian et al., 2013*; *Subramanian et al., 2010*). Flow chambers were built

by applying two strips of double-sided tape to a slide and applying to the coverslip. Sample chamber volumes were approximately 6–8 µL.

Experiments were performed as described previously (*Subramanian et al., 2013*; *Subramanian et al., 2010*). To make antiparallel microtubule bundles, biotinylated microtubules (referred to as 'immobilized microtubules' in text), labeled with rhodamine, were immobilized in a flow chamber by first coating the surface with neutravidin (0.2 mg/ml). Next, 0.2 nM unlabeled PRC1 in BRB80 + 5% sucrose was flushed into the flow chamber. Finally, non-biotinylated microtubules (referred to as moving microtubules) were flushed in the flow cell and incubated for 10–15 min to allow antiparallel overlap formation with the PRC1-decorated immobilized microtubules. To visualize microtubule sliding, PRC1 and Kif4A-GFP and 1 mM ATP were flowed into the chamber in assay buffer (BRB80 buffer supplemented with 1 mM TCEP, 0.2 mg/ml k-casein, 20 µM taxol, 40 mg/ml glucose oxidase, 35 mg/ml glucose catalase, 0.5% b-mercaptoethanol, 5% sucrose and 1 mM ATP) and a time-lapse sequence of images was immediately acquired at a rate of 3 frames/s. Data were collected for 10–15 min. Key experiments and analysis were also performed with GFP-PRC1 and non-fluorescent Kif4A to rule out the effect of GFP on microtubule sliding and stalling by PRC1 and Kif4A. Sliding experiments with Kif4A were performed using the same method.

All experiments were performed on Nikon Ti-E inverted microscope with a Ti-ND6-PFS perfect focus system equipped with a APO TIRF 100x oil/1.49 DIC objective (Nikon). The microscope was outfitted with a Nikon-encoded x-y motorized stage and a piezo z-stage, an sCMOS camera (Andor Zyla 4.2), and two-color TIRF imaging optics (Lasers: 488 nm and 561 nm; Filters: Dual Band 488/561 TIRF exciter). Rhodamine-labeled microtubules and GFP-labeled proteins (either PRC1 or Kif4A) in microtubule sliding assays were imaged by sequentially switching between 488 and 561 channels.

## Single molecule analysis

To visualize single molecules on microtubules, we first immobilized biotinylated microtubule as described above. We then imaged single Kif4A molecules (200 pM Kif4A-GFP and 10 mM AMPMNP) and analyzed the fluorescence intensity distribution.

In order to estimate the total number of molecules in the untagged overlap (both crosslinking and passengers), we measured the fluorescence intensity of microtubule-immobilized single Kif4A-GFP molecules and estimated the number of molecules in the overlap from sliding experiments with 0.2 nM PRC1 + 6 nM Kif4A. Our analyses show that there are on average ~10 molecules/µm the overlap (*Figure 8—figure supplement 2*).

## Image analysis

ImageJ (NIH) was used to process the image files. Briefly, raw time-lapse images were converted to tiff files. A rolling ball radius background subtraction of 50 pixels was applied to distinguish the features in the images more clearly. From these images, individual microtubules sliding events were picked and converted to kymographs by the MultipleOverlay and MultipleKymograph plug-ins (J. Reitdorf and A. Seitz; https://www.embl.de/eamnet/html/body_kymograph.html). The following criteria were used to exclude events from the analysis: (1) Only kymographs where we could confidently identify exactly two microtubules in the bundle were examined further (except for the data in *Figure 7*); (2) Sliding microtubules that encounter another bundle were excluded; (3) Pairs of microtubules with proximal plus-ends at initial time points could not be analyzed due to the very short duration of sliding; (4) For the sliding velocity versus initial overlap analysis, we only included kymographs where the initial overlap and the moving end-tag edge could be clearly distinguished (*Figure 4*); (5) For the microtubule length versus final overlap analysis, we picked kymographs both the immobilized and the moving microtubule edges could be distinguished (*Figure 5*); and (6) In *Figure 4A–D*, we excluded data for initial overlaps greater than 5 µm because of the existence of a few data points.

These kymographs were then further analyzed using a custom MATLAB program (*Wijeratne, 2018*; copy archived at https://github.com/elifesciences-publications/velocity_mt). The program first reads the input image and converts it to an array of intensity values. Next, using the 'bwboundaries' function, the high-intensity edges of the GFP channel kymograph were detected. If the features of the kymograph were clear, the edges of the immobilized end-tag and the moving end-tag were detected. Finally, any repeating elements due to a large amount of noise and poor

contrast were removed by using the 'unique' function. The lines were then converted to $x$, $y$ coordinates at each time point. For unclear MT or GFP channel kymographs, the kymographs can be further processed by ImageJ using the 'Find Connected Regions' plug-in (M. Longair; http://imagej.net/Find_Connected_Regions) to distinguish the features in the kymographs more clearly. This function separates regions in the kymograph based on criteria such as having the same intensity value for the detection of edges. Afterwards, these processed kymographs can be read through the MATLAB program as described above.

To calculate the sliding velocity, the derivative of the position versus time coordinates of the external edge of the moving microtubule end-tag from the GFP channel kymograph was taken. The initial overlap length is calculated from the first time point imaged after flowing in the final reaction mix in our experiment ($t = 0$; example first panel in *Figure 1B*). The initial overlap length, overlap length ($L_{overlap}$) and untagged overlap length ($L_{untagged}$) were measured from the MT channel. The final overlap length ($L_{FO}$) was measured from the MT channel when the end-tags have collided and reached a steady-state. The sum of the end-tag lengths, $L_{ET1}$ and $L_{ET2}$, were determined by measuring the end-tag length before the collision of the end-tags from the GFP channel. The sum of the microtubule lengths, $M_{L1}$ and $M_{L2}$, were measured from the rhodamine-labeled MT channel, and their sum was also plotted.

The overlap intensity ($I_{overlap}$) and untagged overlap intensity ($I_{untagged}$) were measured from the GFP channel. The $\rho_{untagged}$ was determined by $I_{untagged}/L_{untagged}$. The sum of the end-tag intensities, $I_{ET1}$ and $I_{ET2}$, were determined by measuring the end-tag intensity before the collision of the end-tags from the GFP channel. The sliding velocity and initial overlap length at equilibrium were determined at the time point when $I_{overlap}$ is constant.

The initial angle of attachment in the Kif4A control experiments is calculated using the 'Angle' tool in ImageJ. The average end-tag intensity and the average velocity of the end-tag were calculated using the 'TrackMate' plugin (https://imagej.net/TrackMate) in ImageJ.

## Acknowledgements

The authors would like to thank Tarun Kapoor (Rockefeller Univ., USA) for generous support during initial stages of this project and feedback on the manuscript. The authors would also like to thank Yuta Shimamoto (NIG, Japan), Scott Forth (RPI, USA), Meredith Betterton (Univ. of Colorado Boulder, USA) and Doug Martin (Lawrence College, USA) for helpful comments.

## Additional information

### Funding

| Funder | Grant reference number | Author |
|---|---|---|
| Pew Charitable Trusts | | Radhika Subramanian |
| Richard and Susan Smith Family Foundation | | Radhika Subramanian |
| National Institutes of Health | 1DP2GM126894 | Radhika Subramanian |

The funders had no role in study design, data collection and interpretation, or the decision to submit the work for publication.

### Author contributions

Sithara Wijeratne, Data curation, Software, Formal analysis, Investigation, Methodology, Writing—original draft, Writing—review and editing; Radhika Subramanian, Conceptualization, Data curation, Funding acquisition, Writing—original draft, Writing—review and editing

### Author ORCIDs

Sithara Wijeratne  http://orcid.org/0000-0002-3935-926X
Radhika Subramanian  http://orcid.org/0000-0002-3011-9403

Decision letter and Author response

Decision letter https://doi.org/10.7554/eLife.32595.041

Author response https://doi.org/10.7554/eLife.32595.042

## Additional files

### Supplementary files

• Transparent reporting form

DOI: https://doi.org/10.7554/eLife.32595.033

### Data availability

All data generated or analysed during this study are included in the manuscript and supporting files. The underlying source data are also provided.

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
