## [Decision Letter]

Thank you for submitting your article "Geometry of antiparallel microtubule bundles regulates relative sliding and stalling by PRC1 and Kif4A" for consideration by *eLife*. Your article has been reviewed by Anna Akhmanova as the Senior Editor, a Reviewing Editor, and three reviewers.

The reviewers have discussed the reviews with one another and the Reviewing Editor has drafted this decision.

Summary:

The reviewers find that the manuscript addresses an interesting question and presents exciting new results of how motorized sliding of microtubules (by KIF4) is regulated by passive cross-linkers (PRC1) in the case when the motor and the cross-linker directly interact. This topic is important for understanding spindle formation and chromosome segregation.

KIF4 is a multifunctional mitotic kinesin, which localises to the chromosomes during early stages of mitosis and, during anaphase, moves to the spindle midzone through its interaction with PRC1, a microtubule bundling protein, and plays crucial roles in organization of the anti-parallel microtubule bundles. In addition to the motor activity, it has a unique activity to suppress the dynamics of the plus-end of microtubules. Consistent with the latter activity, in vivo depletion phenotypes indicate that KIF4 is important for limiting the anaphase spindle elongation and keeping the central anti-parallel overlap compact. In a previous paper, Subramanian et al., reported on the crystal structures of PRC1 and the formation of high-density regions of KIF4-PRC1 at the microtubule plus-tips, referred to as end-tags. Now it is shown, in a series of nice in vitro experiments (and some modelling), that, as the microtubules slide apart, the tags get closer to each other while the sliding slows down. Sliding stops and stable overlaps are formed when the tags collide. Interestingly, it is observed that structural properties of the initial array regulate PRC1-Kif4A mediated microtubule organization: The sliding velocity scales with the initial microtubule-overlap length and the width of the final overlap scales with the microtubule lengths. Based on these findings, it is suggested that micron-scale geometrical features of antiparallel microtubules can regulate the activity of nanometer-sized proteins to define the structure and mechanics of microtubule-based architectures.

While the reviewers acknowledge that the experiments performed seem very sound (and quite difficult) and that the result in this manuscript provides important new insight, they also raise the concern, whether the progress presented in the current version is large and solid enough to warrant another high impact story (several important ones, including Subramanian et al., 2013, partly by the same authors, have appeared in Cell, Nature Cell Biology). Moreover, the underlying mechanisms (represented by the not yet fully connected experimental and theoretical results) do not seem to be clear at the moment. The following extensive list of major comments would therefore need to be adequately addressed before the paper can be further considered for publication.

Essential revisions:

- It is shown (surprisingly), that the velocity of the KIF4-PRC1 driven sliding scales with the initial overlap length, suggesting that it is dependent on the (untagged?) motor number. This is however not quantified, even though the authors have access to the fluorescence intensities of KIF4-PRC1 (corresponding to motor numbers) in their TIRF data. Please analyze and discuss.

- Similarly, although the motor density is an important parameter (sliding stops at very high density in the "tags"), it is at the moment unclear how does the sliding velocity scale with the density of the motors. Only two stopping events are quantified (Figure 3H and Figure 4—figure supplement 1P) suggesting that it might also be the increase of the density of the untagged motors causing the slowdown. Please analyse from more events and discuss.

- To address the slow-down mechanism, the authors are advised to investigate the sliding of fully overlapping microtubules, in which they can generate different KIF4-PRC1 densities/numbers by different KIF4-PRC1 concentrations in solution. Assembling the microtubule sandwiches for these experiments in ADP and then exchanging ADP to ATP would allow the authors to initially measure the sliding velocities at different KIF4-PRC1 densities homogenously distributed in the overlap (without the presence of the end-tags) and then to observe the change of the velocities during the end-tag formation. Please add such data, analyse and discuss.

- It is somewhat dissatisfying that there is no direct correlation of individual PRC1 and Kif4A complexes. What is the lifetime of the attachment of Kif4A to PRC1? Should one see it as 1:1 complexes that live forever, or is binding more intermittent and should the interaction be seen more as a collective effect (many PRC1s with each Kif4A, at a given moment or after each other)? This could e.g. have important consequences for the mechanism of action. Please discuss.

- KIF4 interacts with PRC1 via its C-terminal tail domain. However, the same domain also has a microtubule-binding activity independent of PRC1 (Bieling, 2010, Figure 7—figure supplement 1). Although it is unclear how strong this interaction is, considering that both the interactions between PRC1 and KIF4 and between PRC1 and unbundled microtubules are weak (K_d_ ~0.3 μm (Subramanian, 2013) and ~0.6 μm (Bieling, 2010), respectively), it is possible that the KIF4 tail-microtubule interaction might also play an important role in the authors' experimental system. Please perform control experiments and/or discuss.

- The relatively low affinity between KIF4 and PRC1 (~0.3 μM, cf. K_d_~8 nM between PRC1 and anti-parallel microtubule bundle) raises the question of what percentages of KIF4 and PRC1 in the tagged and untagged overlaps are in the complex. This might not be important if there is no productive force generation by KIF4 molecules not complexed with PRC1. However, considering the presence of the second microtubule-binding site of the KIF4 tail (see above), it is important to clarify the states of the PRC1-KIF4 complex formation. Reduction of the sliding velocity by increased PRC1 and the reversal by additional KIF4A is consistent with a picture that PRC1 holds the anti-parallel overlap of MTs and KIF4A, which is not in a complex with PRC1, slides them. Please discuss.

- Which fraction of KIF4-PRC1 is driving the sliding and what is causing the slow-down (during the phase-2). Is the decrease of velocity in phase-2 due to the decrease in the number of "untagged" KIF4-PRC1 or an increase in their density? Does friction play any role? Please discuss.

- The description of the result from subsection “Characterization of relative microtubule sliding in the PRC1-Kif4A system” (density and intensity of Kif4A in the different overlap regions) is not very clear. What is meant with "retained"? What is meant by the final sentence of this part? Are these statements based on only two example kymographs (Figure 3E and Supplementary figure 1C-L). For Figure 3H it appears that the overlap keeps on increasing until the end of the experiment (unlike the event in Supplementary figure 1C-L). Please explain more clearly.

- To what extend is the analysis of the end tag lengths (Figure 4AB) affected by the diffraction limited imaging used? How different are the end-tagging lengths on single microtubules and in microtubule overlaps? Please discuss.

- In their theoretical model, the authors implicitly assume that one of the two PRC1 ends that binds the same microtubule with the one that KIF4 interacts with (i.e. the bottom one in Appendix Figure 1A) doesn't interfere with the MT sliding. It is not clear how this is justified. It would be more natural that one end of a PRC1 dimer (upper one in Appendix Figure 1A) works as a supporting point for productive sliding while the other end (bottom one) works as a drag against the stepping of KIF4. Please discuss and resolve the issue.

- On one hand, modeling the PRC1-MT interaction as a slipping tether using the formulation by Grover et al., is an excellent idea. However, the model presented by the authors is not properly describing their experimental conditions. Their model is for a different situation in which KIF4 is tethered on the PRC1 bridging two microtubules (one of them is immobilized) and it drives the sliding of the third, non-immobilized microtubule. Thus, it is not surprising that there is a big inconsistency between the model calculation and the experimental measurements in the order of the sliding velocity. Please resolve.

- On the other hand: Is it really warranted to use a similar model to diffusion (and drag) in a membrane for diffusion of the PRC1 complexes over the MT? PRC1 binds to the MT, most likely with 8 nm periodicity, with relatively large barriers / wells; in the membrane stuff is much more continuous. Would in the current situation such a view not be too simplistic, e.g. not taking into account 'non-linear' / 'out-of-equilibrium' effects due to motor action (i.e. increased loads on the motor which could result in changes in motor action (velocity, release) including a non-linear scaling of friction with motor number (see Lansky et al., 2015).

- According to the authors' formula in the Appendix, 𝑣_𝑀𝑇_ is hyperbolic against a dimensionless value x=a∙d∙l/L_MT_ ("Michaels-Menten" type, passing origin and approaching max value = 𝑣*_step_*). A characteristic parameter is f∙δ, which corresponds to x which gives a half maximal 𝑣_𝑀𝑇_ (equivalent to K_M_ for "Michaels-Menten"). In a regime where x is below this, the 𝑣_𝑀𝑇_ ~ x relationship becomes nearly linear. However, calculations with the values in Table A1 result in f∙δ = 1.72 x 10^-5. This is too small, and it is impossible to make x smaller than this by any realistic combinations of a, d, l and *L_MT_*. In other words, 𝑣_𝑀𝑇_ is almost equal to 𝑣*_step_* irrespective of l, the lengths of overlap. Please explain how the curves in Appendix Figure 1 were drawn. The actual values of a, d and D used should be presented.

- The modeling should be connected closer to the experiments. For example, a∙d is essentially the line density of PRC1-Kif4A in the overlap and thus should be measurable. Then, the actual D should be able to be determined with actual x and 𝑣_𝑀𝑇_ measurements using the authors' formula and can be compared with reported values. Moreover, it would be helpful to indicate what trends (and numbers) belong to the experimentally tested parameter space. The simulations (at least their results) should be discussed more prominently in the main text /Discussion section. Please add this information.

- Biological significance: It is unclear whether the situations studied in this work actually occur in the cell. Although the end-tagging of astral microtubules near spindle poles by PRC1 was demonstrated in Subramanian et al., 2013, it remains unclear whether KIF4A takes part in these tags. Even so, it is unclear whether there is a cellular situation in which microtubules are first tagged (and stabilized) with PRC1 and KIF4A and then bundled. In general, PRC1 localizes on the metaphase spindle although weakly and diffusely while KIF4 is associated with chromosomes before anaphase onset. PRC1 can form midzone bundles without KIF4. Please discuss.

[Editors' note: further revisions were requested prior to acceptance, as described below.]

Thank you for resubmitting your work entitled "Geometry of antiparallel microtubule bundles regulates relative sliding and stalling by PRC1 and Kif4A" for further consideration at *eLife*. Your revised article has been evaluated by Anna Akhmanova (Senior Editor), a Reviewing Editor, and three reviewers.

The involved editors as well as the reviewers acknowledge that you did a great job of seriously considering the earlier comments, including the performance of additional experiments. The conclusions are now much more solid and the context with previous studies and other systems is discussed much more clearly, highlighting why this study is important and exciting. The manuscript has tremendously gained by the revision and it is felt that the work in general is very well suited for *eLife*.

However, there are some remaining issues that need to be addressed before your manuscript can possibly be accepted for publication:

1) Please have a look at the following (conclusion of the Figure 4, subsection “Examining the time-dependent changes during microtubule sliding in the PRC1-Kif4A system”): "…velocity of microtubule sliding.… is determined by.…. the total number of sliding competent molecules in the untagged overlap." Velocity increases with increasing (untagged) motor number (Figure 4A-D) and "The microtubule movement can subsequently proceed at a *constant velocity*, even when the overlap shrinks,…" (i.e. when the untagged *motor number decreases* – phase 1 in the Figure 4F and Figure 4—figure supplement 2) "…possibly through increasing the density of motor molecules during relative sliding." This would mean that increasing the motor density should compensate for decreasing the motor number to keep the velocity constant. That is, velocity should increase with increasing motor density.

However: (i) How an increased motor density would result in an increased sliding velocity is intriguing and the authors should probably comment on this. (ii) In contradiction with their statement, the authors show that (at least in some concentration regime) this is not the case (Figure 4—figure supplement 1F).

2) Connected to (1): An essential statement of the paper is that the sliding velocity scales with the initial overlap length. However, Figure 4E-H shows that the sliding velocity stays constant for a shrinking overlap lengths (called phase 1). How is the "initial overlap length" defined here? What means "initial"? Most surprisingly, the overlap intensity increases during this phase. How is that explained? Is there equilibrium in binding achieved before?

3) With regard to the interaction between the C-terminal tail of KIF4A and microtubules it is stated that the interaction between the KIF4A tail and microtubules are not strong. However, the PAGE image in Figure 2 clearly shows there is some interaction between them. The signals of the bands that correspond to KIF4A (but don't appear in 'PRC1' lanes) in the precipitates (KIF4A(C-term) 'P') increase with the increasing amount of microtubules. This indicates a weak but significant interaction between KIF4A C-tail and microtubules. The authors' statements such as "We observed no significant microtubule-association of this domain" (subsection “Molecular determinants of the sliding and cross-bridging in the PRC1-Kif4A system”) or "the C-terminus of Kif4A does not directly bind microtubules" are thus not true.

This interaction is indeed 'very' weak as a MAP. The dissociation constant might be at the orders of 100 µM or 1 mM. However, it should be kept in mind that this domain is not floating alone in solution but is part of a kinesin-like motor protein, which strongly accumulates at the plus-ends of microtubules. The local concentration of the domain can be the order of 1~10 mM, which seems to be comparable with the weak but significant interaction detected in Figure 2A.

Similarly, hydrodynamics data at the protein concentration of 5 µM might not be strong enough to exclude the possibility that Kif4A might form oligomers at the crowded condition. The second (very weak) binding site on the C-terminal tail or oligomerization seems to be more plausible as an explanation of the MT-sliding by Kif4A alone. At least the data are not strong enough to exclude these possibilities. Please discuss all of the above.

4) The response to comment 13 is unsatisfactory. The difference of units doesn't matter if the calculation is performed with physical quantities (numbers + units).

The characteristic parameter (the initial overlap that gives the half maximum velocity, analogous of K_M_ in the Michaelis-Menten kinetics, here called λ) λ should be at the order of µm or bigger. However, when repeating the calculation with the values provided in the revision, the λ is calculated to be at the order of picometer. Thus, what the authors' theory predicts is that the sliding velocity is independent of the initial overlap length. The theoretical curves in Appendix Figure 1 appear inconsistent with the authors' theory and the parameters provided. Please check.

5) In the calculation, d = 13 and a = 0.4 is assumed – meaning that there should be about 600 molecules active in a 1 micron long MT overlap? Is that a reasonable assumption? Can't an upper bound of the motor number be (roughly) estimated from the fluorescence intensity?

6) With regard to the contribution of the end tags on sliding: Kif4A alone can form an end tag, which drives movement of a non-immobilized microtubule along an immobilized one. It is not clear why the similar end-tag doesn't contribute much to the MT sliding in the Kif4A-PRC1 regime. How fast is the movement driven by Kif4A alone end tags in the experiments represented by Figure 2C and D (can't be currently estimated because scale bars are missing in these panels)?

[Editors' note: further revisions were requested prior to acceptance, as described below.]

Thank you for your second resubmission of your work entitled "Geometry of antiparallel microtubule bundles regulates relative sliding and stalling by PRC1 and Kif4A" for further consideration at *eLife*. Your revised article has been evaluated by Anna Akhmanova (Senior Editor), a Reviewing Editor, and three reviewers.

The involved editors as well as the reviewers acknowledge that you did a great job in addressing the points raised. Removing the earlier modeling part and replacing it by a kind of "duty ratio" discussion makes sense and is a nice way to extract quantitative information out of the data. While this current description is admittedly not as advanced / informative as a real model (attempted in the last version of the manuscript) it nevertheless provides useful mechanistic insight. Given the enormous amount of very high quality experimental data, the taken approach is regarded fine for this paper. Future work could go into a more advanced model. Hence, the manuscript is now in principle regarded suitable for publication in *eLife*.

There is one remaining point that the reviewers and editors find of crucial importance before potential acceptance of the paper: The usage of the term/concept "duty ratio" does not seem to be fully appropriate in the presented context. The traditional/authentic "duty ratio" is about the temporal fraction of the crossbridge cycle (= ATPase cycle) of a single motor head in which it is attached to the filament and makes its working stroke. In contrast, the situations the authors imagine are (i) dissociation of the PRC-Kif4A complex from the microtubule, and (ii) slippage of PRC1 on the MT. These will influence the fraction of ATPase cycles that actually result in the sliding of the non-immobilized microtubule, i.e., the fraction of the productive stepping by Kif4A. What the authors call "duty ratio, f", is a mixture of the authentic duty ratio (as to the crossbridge/ATPase cycle) and the effect of the futile cycles. It is not appropriate to skip these details and call the parameter simply "duty ratio". A better term to describe the scenario may be "sliding efficiency". In any case, the authors should explicitly mention that they mean something slightly, but substantially, different than what "duty ratio" has been used for before.

In other words, both the authentic duty ratio and the fraction of productive stepping would influence the sliding velocity in a similar way, following the same form of a mathematical formula (2) in subsection “Mechanism of overlap-length dependent sliding by PRC1 and Kif4A”, as a first approximation. However, their meanings are quite different. A low duty ratio motor can still be highly energy efficient (like dynein). On the other hand, futile sliding simply wastes energy of ATP hydrolysis as a slippery between PRC1 and MTs or a dissociation between PRC1 and KIF4A. Along these lines: Is the low "duty ratio" of the PRC1-Kif4A complex in MT sliding consistent with its highly processive motility along a MT? A quantitative argument is necessary as to the difference in the loads on the PRC1-Kif4A complex between the two conditions; the MT sliding and the single particle motility. Statements like "…microtubule sliding by the PRC1-Kif4A complex can be considered as microtubule movement driven by an ensemble of low duty-ratio motors,.… " need to be revised accordingly.

---

## [Author Response]

Summary:The reviewers find that the manuscript addresses an interesting question and presents exciting new results of how motorized sliding of microtubules (by KIF4) is regulated by passive cross-linkers (PRC1) in the case when the motor and the cross-linker directly interact. This topic is important for understanding spindle formation and chromosome segregation.KIF4 is a multifunctional mitotic kinesin, which localises to the chromosomes during early stages of mitosis and, during anaphase, moves to the spindle midzone through its interaction with PRC1, a microtubule bundling protein, and plays crucial roles in organization of the anti-parallel microtubule bundles. In addition to the motor activity, it has a unique activity to suppress the dynamics of the plus-end of microtubules. Consistent with the latter activity, in vivo depletion phenotypes indicate that KIF4 is important for limiting the anaphase spindle elongation and keeping the central anti-parallel overlap compact. In a previous paper, Subramanian et al., reported on the crystal structures of PRC1 and the formation of high-density regions of KIF4-PRC1 at the microtubule plus-tips, referred to as end-tags. Now it is shown, in a series of nice in vitro experiments (and some modelling), that, as the microtubules slide apart, the tags get closer to each other while the sliding slows down. Sliding stops and stable overlaps are formed when the tags collide. Interestingly, it is observed that structural properties of the initial array regulate PRC1-Kif4A mediated microtubule organization: The sliding velocity scales with the initial microtubule-overlap length and the width of the final overlap scales with the microtubule lengths. Based on these findings, it is suggested that micron-scale geometrical features of antiparallel microtubules can regulate the activity of nanometer-sized proteins to define the structure and mechanics of microtubule-based architectures.While the reviewers acknowledge that the experiments performed seem very sound (and quite difficult) and that the result in this manuscript provides important new insight, they also raise the concern, whether the progress presented in the current version is large and solid enough to warrant another high impact story (several important ones, including Subramanian et al., 2013, partly by the same authors, have appeared in Cell, Nature Cell Biology). Moreover, the underlying mechanisms (represented by the not yet fully connected experimental and theoretical results) do not seem to be clear at the moment. The following extensive list of major comments would therefore need to be adequately addressed before the paper can be further considered for publication.

In recent years, in vitro reconstitution-based studies have revealed the mechanisms by which the collective activities of motor and non-motor proteins regulate the architecture of microtubule arrays. However, there are relatively few studies on how the micron-scale geometrical features of filament arrays, in turn, tune the output of microtubule associated proteins [Braun et al., 2011; Braun at el., 2017; Bieling, Telley and Surrey, 2010; Shimamoto, Forth and Kapoor, 2015]. Our study uncovers new mechanisms by which the geometrical features of antiparallel microtubule bundles, such as filament length and overlap width, can be ‘read’ by associated proteins to regulate the dynamics, stability and architecture of cytoskeletal structures.

While previous studies have characterized the molecular interactions and regulation of microtubule dynamics by PRC1 and Kif4A, microtubule sliding by this protein module is poorly characterized [Bieling, Telley and Surrey, 2010; Nunes et al., 2013; Subramanian et al., 2013]. This study reveals unexpected emergent features of microtubule organization by a protein module composed of a pair of interacting motor and MAP: (i) Microtubule sliding and stalling by PRC1 and Kif4A results in the organization of a stable bundle in which the plus-ends of microtubules are closely aligned. (ii) The size of the final stable antiparallel overlap scales with the lengths of the crosslinked microtubules. Therefore, in addition to preventing microtubules from sliding apart, PRC1-Kif4A end-tags act as ‘molecular rulers’ to define the width of the stable antiparallel overlap. (iii) The microtubule sliding velocity in the PRC1-Kif4A system depends on the initial length of the antiparallel overlap. This is unusual for microtubule movement driven by an ensemble of motor proteins. We have reorganized the discussion to emphasize these findings.

Microtubule sliding and stalling has been described in another motor-MAP system (pombe Ase1 and kinesin-14 Ncd), and more recently with kinesin-14 HSET [Braun et al., 2011; Braun at el., 2017]. While the end result is the formation of a stable antiparallel overlap in all three systems, there are fundamental mechanistic differences between the results here and the published work. (i) In both the published studies, the sliding velocity is not determined by the initial microtubule overlap length as in the PRC1-Kif4A system, but rather changes continuously as the overlap shrinks (i.e. adaptive rather than determined by the initial state). (ii) In both the Ncd-Ase1 and the HSET systems, it is proposed that the reduction in velocity is due to an increase in forces opposing motor stepping as the overlap shrinks. Such a force-balance mechanism cannot account for initial length-dependent sliding in our system, and we discuss an alternative mechanism consistent with our data. (iii) A force balance mechanism also underlies the establishment of a stable overlap in the published studies. We propose an alternative mechanism consistent with our findings. (iv) Scaling of final overlap length with the sum of microtubule lengths is not an inherent feature of the Ase1-Ncd and the HSET systems. Overall, we think our study represents a different class of mechanism for both the formation of a stable array as well as overlap length-dependent sliding.

We have now performed additional experiments, analyses and extended the discussion to clarify the mechanisms underlying microtubule sliding and stalling in this system. Briefly, our main conclusions are: (i) Length-dependent sliding arises from dynamically associating PRC1-Kif4A molecules in the untagged region of the overlap. We discuss the ‘slipping tether model’ as a simple mechanism that is consistent with our data. (ii) Slowdown is due to motors entering the high-density regime proximal to end-tag and the movement stalls when the end-tags merge and stepping by the motor protein is suppressed. We discuss steric hindrance as the predominant mechanism underlying both slow down and stall in this system.

Essential revisions:1) It is shown (surprisingly), that the velocity of the KIF4-PRC1 driven sliding scales with the initial overlap length, suggesting that it is dependent on the (untagged?) motor number. This is however not quantified, even though the authors have access to the fluorescence intensities of KIF4-PRC1 (corresponding to motor numbers) in their TIRF data. Please analyze and discuss.

As per the reviewer’s suggestion, we are able to quantify fluorescence intensity in the untagged and end-tagged regions of the overlap. Assuming that the number of sliding molecules is proportional to the total number of molecules, our analyses suggest that length dependent sliding arises from molecules in the untagged overlap region. The following data have now been added to the manuscript:

Correlation between overlap length and intensity: We find that the fluorescence intensity of Kif4A-GFP is proportional to total overlap length (Figure 4—figure supplement 1C). From this, we infer that the sliding velocity scales with the number of motor molecules in the overlap. We prefer to plot length rather than intensity because it is overall a less noisy measurement and is not impacted by illumination heterogeneity across the field of view or photobleaching. We have now included a plot showing the correlation (N = 20; Pearson’s correlation coefficient = 0.90) between overlap length and fluorescence intensity of a randomly chosen subset of the data to the Supplement (Figure 4—figure supplement 1C).

In addition, we have now included the following data, which reveal that sliding velocity scales with the untagged overlap length/intensity and not the end-tagged length/intensity:

Untagged overlap vs. sliding velocity (Figure 4—figure supplement 1D-E). The data show a correlation between untagged overlap length and sliding velocity under all conditions where we observe a correlation between total overlap length and sliding velocity. Assay conditions: (i) 0.2 nM PRC1 and 6 nM Kif4A-GFP (Figure 4—figure supplement 1D; black; N = 64; Pearson’s correlation coefficient = 0.54), (ii) 0.5 nM GFP-PRC1 and 6 nM Kif4A (Figure 4—figure supplement 1E; red; N = 20; Pearson’s correlation coefficient = 0.83), and (iii) 1 nM GFP-PRC1 with 12 nM Kif4A (Figure 4—figure supplement 1E; blue; N=22; Pearson’s correlation coefficient= 0.79). We also see a correlation between untagged overlap length and Kif4A-GFP intensity (Figure 4—figure supplement 1D inset). This suggests that the number of motor molecules in the untagged overlap is responsible for length-dependent sliding in this system.

End-tagged overlap vs. sliding velocity data: In the case of end-tags, the intensity is a more reliable measurement than length, as some of the end-tags are very short (less than 5 pixels). Analysis of both end-tag length and intensity as a function of sliding velocity indicate that there is no positive correlation between the number of molecules at the end-tag and the sliding velocity (Figure 4—figure supplement 1A-B, Pearson’s correlation coefficient = between -0.18 and -0.40).

Note: we would like to emphasize that while intensity measurements reflect the total number of molecules in the overlap, measuring the precise number of crosslinking motor proteins within a bundle that contribute to microtubule sliding is challenging for the following reasons. First, fluorescence intensity represents both the Kif4A molecules that are walking on individual microtubules and those that participate in crosslinking and sliding microtubules [Shimamoto, Forth and Kapoor, 2015]. We expect that the number of crosslinking PRC1-Kif4A molecules will be fewer than the number of passenger molecules. This is because PRC1 molecules cross-bridge over a narrow intermicrotubule distance range [Subramanian et al., 2010]. Second, PRC1 and Kif4A do not form a high-affinity stable complex, and sliding is likely sustained by molecules that are dynamically dissociating and reassociating within the bundle (further elaborated in response to comment #4).

2) Similarly, although the motor density is an important parameter (sliding stops at very high density in the "tags"), it is at the moment unclear how does the sliding velocity scale with the density of the motors. Only two stopping events are quantified (Figure 3H and Figure 4—figure supplement 1P) suggesting that it might also be the increase of the density of the untagged motors causing the slowdown. Please analyse from more events and discuss.

We have now included an additional panel of 4 sliding and stalling events along with quantification in the supplement (see Figure 4—figure supplement 2).

How does sliding velocity in phase-1 scale with motor density (as recently reported for HSET)? We find no correlation between sliding velocity and the average untagged overlap motor density in phase-1 for a dataset in which we see a positive correlation between sliding velocity and untagged overlap length/intensity [0.2 nM PRC1 and 6 nM Kif4A-GFP] (Pearson’s coefficient: -0.15; N = 20). This suggests that in this regime, the velocity depends on the number but not the density of motor molecules (Figure 4—figure supplement 1F).

Is the transition from phase-1 to phase-2 due to increase in density of untagged motors?

We analyzed kymographs from experiments conducted with 0.2 nM PRC1 and 6 nM Kif4AGFP and examined the individual Kif4A-GFP density versus time profiles (representative data in Figure 4H and Figure 4—figure supplement 1P and Figure 4—figure supplement 2D, 2H, 2L, 2P). We find that the Kif4A-GFP density in the untagged overlap increases during both phase-1 and phase-2 of movement. So, the transition from phase-1 (constant velocity sliding) to phase-2 (slowdown) does not correspond to a switch from constant to increasing density.

Next, we considered the alternative possibility that the transition from sliding to stalling coincides with the end-tags on the moving and immobilized microtubules arriving at close proximity. At this high-density region, microtubule sliding would first slow down before complete stall is reached. We noticed that in ~50% of the events (47/98; 0.2 nM PRC1 + 6 nM Kif4A-GFP), the transition from phase 1 to 2 occurs when the end-tags have nearly merged and our image analysis algorithm does not resolve the two end-tags (examples: see kymographs in Figure 4E, Figure 4—figure supplements 2I, 2M). We reasoned that if high protein concentration proximal to end-tags is the reason for slow-down, the protein density at the phase-1 to phase-2 transition should be similar under different experimental conditions. Kif4A-GFP density measurement and fluorescence line-scan analysis at the phase-1 to phase-2 transition timepoints in two experimental conditions show that this is indeed the case (Figure 4—figure supplement 4A-B). Furthermore, comparison of the phase-1 to phase-2 transition-point intensity with the average end-tag intensity at the same time point suggests that on average slowdown occurs when the intensity is >70% of the end-tag intensity (Figure 4—figure supplement 4B-C). These observations suggest that sliding slows down proximal to the end-tags, when the untagged overlap is short and the motors encounter a high protein-density.

Together, these analyses suggest that at the (i) phase-1: sliding depends on initial overlap length but not overlap density under the same protein concentrations (ii) phase-1 to -2 transition: sliding slows down due to high protein density. We discuss steric inhibition to stepping as the mechanism underlying slowdown and stall in this system (discussed in response to comment #10 and in the main text under the section ‘Mechanism for transition from constant velocity sliding to slow-down and stalling in the PRC1-Kif4A system’ and Discussion section).

3) To address the slow-down mechanism, the authors are advised to investigate the sliding of fully overlapping microtubules, in which they can generate different KIF4-PRC1 densities/numbers by different KIF4-PRC1 concentrations in solution. Assembling the microtubule sandwiches for these experiments in ADP and then exchanging ADP to ATP would allow the authors to initially measure the sliding velocities at different KIF4-PRC1 densities homogenously distributed in the overlap (without the presence of the end-tags) and then to observe the change of the velocities during the end-tag formation. Please add such data, analyse and discuss.

We have now performed the experiment suggested above and reanalyzed our previous data to address how the sliding velocities change with end-tag formation. Based on the analyses, we conclude that the initial overlap-length dependent sliding is independent of end-tag length. We discuss the findings below:

As suggested by the reviewers, microtubule bundles were assembled with protein and ADP (0.2 nM GFP-PRC1 + 6 nM Kif4A and 2 mM ADP; experimental details in the legend). Under these conditions, no end-tags are observed. Upon exchanging the buffer with ATP (no additional protein in solution), we observe concurrent sliding and end-tag formation even at the earliest time point we could image (Author response image 1). We have also performed this experiment with Kif4A and GFP-PRC1 and obtain the same results. This is expected because the rate of PRC1-Kif4A movement on single microtubules (500 nm/s), which leads to end-tag formation, is faster than the microtubule sliding velocity [Bieling, Telley, and Su, 2010; Subramanian, 2013]. Hence, we cannot measure sliding velocities at homogeneous Kif4A-PRC1 densities by this method. As such, experimentally decoupling end-tag formation and sliding is not feasible. [Note: since solution PRC1/Kif4A is washed out during the ATP exchange, the number of molecules in the overlap is reduced and the sliding velocity is much slower. It is therefore difficult to assess the over-lap-length dependence of sliding under these conditions.] As this experiment is nearly identical to the experiments in Figure 6 of the main text and does not provide additional mechanistic information, we have not included it in the revised manuscript. Instead, we have addressed the question of how velocities change during the end-tag formation through additional analyses of our data as described below.

**Author response image 1. respfig1:** Method:Biotinylated rhodamine-labeled microtubules were immobilized in a flow chamber. 0.2 nM un-labeled PRC1 in assay buffer was flushed into the flow chamber. Next, non-biotinylated microtubules were flushed in the flow cell and incubated for 10-15 mins to allow antiparallel overlap formation. Afterwards, 6 nM Kif4A-GFP and 2 mM ADP were flowed into the chamber. To visualize microtubule sliding, 2 mM ATP were flowed into the chamber in assay buffer and a time-lapse sequence of images was immediately acquired at a rate of 5 frames/s for 10-15 min.

(A) Kymograph shows the relative sliding and stalling of antiparallel microtubules (red, microtubules; green, Kif4A-GFP). Assay condition: 0.2 nM PRC1 + 6 nM Kif4A-GFP + 2 mM ADP. Scale bar: *x*: 1 µm and *y*: 1 min. (B) Time record of the instantaneous sliding velocity of the moving microtubule derived from the kymograph in (A). The dashed lines demarcate the three phases observed in the sliding velocity profile: (i) constant phase, (ii) slow down and (iii) stalling. (C) Time record of the total overlap length (red; 𝐿_𝑜𝑣𝑒𝑟𝑙𝑎𝑝_) derived from the kymograph in (A). (D) Total fluorescence intensity (dashed gray; 𝐼_𝑜𝑣𝑒𝑟𝑙𝑎𝑝_), total fluorescence intensity in the untagged region of the overlap (solid purple; 𝐼_𝑢𝑛𝑡𝑎𝑔𝑔𝑒𝑑_), and fluorescence density in the untagged region of the overlap (solid green; ρ_𝑢𝑛𝑡𝑎𝑔𝑔𝑒𝑑_) derived from the kymograph in (A). (E-H) Another representative kymograph from this experiment. Panels E-H are equivalent to panels A-D. (I) Histogram of the average sliding velocity calculated from the constant velocity movement in phase-1. Assay condition: 0.2 nM PRC1 + 6 nM Kif4A-GFP + 2 mM ADP (mean:8 ± 5 nm/s; N=16).</Author response image 1 title/legend>

In order to address the question of whether either the sliding velocity in phase-1 or the transition from phase-1 to phase-2 is correlated with end-tag formation or length, we reexamined our data.

We examined if there was a correlation between end-tag length and average sliding velocity in phase-1. For this analysis, we calculated the end-tag length for phase-1 movement in two different ways (end-tag length at t = 0 and the time point before end-tags collide). No significant positive correlation was observed between end-tag length/intensity and sliding velocity (Figure 4—figure supplement 1A-B; correlation coefficients between -0.18 and -0.40). These results suggest that phase-1 sliding velocity is independent of the end-tag length, instead scales with the untagged overlap length (Figure 4—figure supplement 1D-E).

We analyzed individual events in which we could clearly observe an increase in end-tag length during sliding (the end-tag establishment phase; example: Figure 4—figure supplement 1G-I). We see no significant correlation between instantaneous velocity and increase in end-tag intensity during sliding in 10 events analyzed during phase-1 movement or at the transition from phase-1 to phase-2 (Author response image 2, black arrows indicate phase-1 to -2 transition).

Together these analyses suggest that both phase-1 sliding velocity and the transition from phase-1 to phase-2 are independent of end-tag length (also discussed in response to comment #7).

**Author response image 2. respfig2:** Instantaneous sliding velocity versus moving end-tag intensity, 𝐼_𝐸𝑇2_. (**A**) Instantaneous sliding velocity versus moving end-tag intensity, 𝐼_𝐸𝑇2_, plots from 16 kymographs (colored circles). (**B**) Zoomed-in view of three events in (**A**). The arrows indicate the phase 1-2 transition point.

4) It is somewhat dissatisfying that there is no direct correlation of individual PRC1 and Kif4A complexes. What is the lifetime of the attachment of Kif4A to PRC1? Should one see it as 1:1 complexes that live forever, or is binding more intermittent and should the interaction be seen more as a collective effect (many PRC1s with each Kif4A, at a given moment or after each other)? This could e.g. have important consequences for the mechanism of action. Please discuss.

This is an excellent question and we think that it does impact the mechanism. The binding affinity between PRC1 and Kif4A is 0.3 μM. Consistent with this, we cannot isolate stable PRC1-Kif4A complexes using size-exclusion chromatography at sub-μM concentrations or observe single particles of the complex on an EM grid. Based on our previous structural data and preliminary chromatography results, the interaction appears to be stoichiometric (unpublished observations). Therefore, under our experimental conditions, PRC1 and Kif4A are not likely to act as stable complexes. However, prior work shows that the lifetime of Kif4A is increased on microtubules in the presence of PRC1 [Bieling, Telley and Surrey, 2010; Subramanian et al., 2013]. This suggests that while the inherent affinity between PRC1 and Kif4A is not high and the estimated dissociation rate is on the order of a few seconds, the localization of these proteins to microtubules results in high rates of re-association. Together, these observations suggest that the interaction between these proteins is best described as a series of dissociation and re-association events of Kif4A with the same or neighboring PRC1 molecule (note: similar to the proposal in Ref. Bieling, Telley and Surrey, 2010).

This result has an important consequence and we briefly mentioned it in the Discussion section of the original submission. We now state it more clearly in the revised submission. Briefly, if PRC1 and Kif4A form a stable high-affinity complex then it would essentially function as single crosslinking molecule where motor stepping is tightly coupled to sliding. Instead, in this system, because of dissociation, every step by the motor may not lead to a corresponding movement of the transport microtubule. The increased decoupling will result in increased dependence of microtubule sliding velocity on motor number. Essentially, even though Kif4A is a high duty ratio motor with respect to its stepping behavior on single microtubules, the PRC1-Kif4A complex may act as a low duty ratio complex with respect to microtubule sliding due to the decoupling of motor stepping from microtubule sliding.

5) KIF4 interacts with PRC1 via its C-terminal tail domain. However, the same domain also has a microtubule-binding activity independent of PRC1 (Bieling, 2010, Figure S4). Although it is unclear how strong this interaction is, considering that both the interactions between PRC1 and KIF4 and between PRC1 and unbundled microtubules are weak (Kd ~0.3 μm (Subramanian, 2013) and ~0.6 μm (Bieling, 2010), respectively), it is possible that the KIF4 tail-microtubule interaction might also play an important role in the authors' experimental system. Please perform control experiments and/or discuss.

In the previous reported work [Bieling, Telley and Surrey, 2010], it is stated that full-length *Xenopus Laevis* Kif4 did not mediate microtubule bundling in ATP (bundling was only observed with AMPPNP). Similarly, we are also unable to establish aligned bundles with Kif4A in the presence of 1 mM ATP. We see end-tags forming on microtubules but these do not form antiparallel bundles. It is difficult to interpret the microtubule-binding activity of the C-terminus of Kif4A from indirect bundling experiments at high motor concentrations (especially under tight binding conditions such as in the presence of AMPPNP). Therefore, we directly examined microtubule binding by the C-terminus of Kif4A using co-sedimentation assays (Figure 2A).

We expressed and purified a construct (aa 733-1232) that comprises of the globular C-terminus domain of Kif4A (‘tail’; aa 1000-1232) and the coiled coil stalk (aa 732-999; note: we were unable to purify a longer construct due to problems with protein aggregation). This construct contains both the PRC1 and the DNA binding domains of Kif4A, and the globular tail domain is an attractive candidate as a microtubule-binding domain based on the presence of similar domains in other kinesins. However, our results show negligible sedimentation of this construct (< 5%) even in the presence of 8 μM tubulin (Figure 2A). Therefore, it is unlikely that Kif4A molecules crosslink microtubules using the C-terminus domain.

To examine alternative mechanisms of crosslinking by Kif4A, we bound Kif4A-GFP to immobilized microtubules at low ATP concentrations (10 nM). Under these conditions, we do not see end-tag formation and the protein is bound all along the microtubules. However, we observe a few pairs of microtubules interacting at random orientations. We then flow in 1 mM ATP and Kif4A-GFP, which result in end-tag formation on both microtubules. If the end-tag on the nonbiotinylated microtubule can contact the immobilized microtubule, it results in tip-mediated gliding of one microtubule over the other. Interestingly, the angle of attachment is usually low (0-30 degrees), and the movement halts when the end-tags collide (Figure 2C-E). We see no dependence of sliding velocity on Kif4A end-tag intensity (Figure 4—figure supplement 1Q). This result is consistent with our analyses for end-tag intensity versus sliding velocity in the PRC1Kif4A experiments in Figure 4—figure supplement 1A.

These findings indicate that the Kif4A-dense microtubule tip can slide along another microtubule even though the C-terminus of Kif4A does not bind microtubules. How might this be possible? Previous EM studies have shown that under crowded conditions, kinesins adopt a one-head bound conformation [Hirose et al., 2000; Hoenger et al., 2000]. Therefore, it is possible that the end-tags have an array of detached motor domains particularly at the very tip, and the collective activity of single heads could drive microtubule movement. Alternatively, it is possible that at the crowded tip, Kif4A molecules could oligomerize to form tetramers – however, even when concentrated to >5 μM, Kif4A does not elute as oligomers in size exclusion chromatography. We therefore favor the first model at this time. Interestingly, a similar mechanism of aster formation by dynein-dynactin has been suggested recently [Tan et al., 2018].

Currently, it remains unclear if in aligned antiparallel bundles (180 degrees), Kif4A molecules at the end-tags can similarly contribute to sliding. Regardless, based on analyses of sliding velocity as a function of end-tag intensity in both the Kif4A and the PRC1+Kif4A experiments (Figure 4—figure supplement 1A and 1Q), it is clear that movement driven by end-tagged motors does not significantly contribute to the initial overlap length dependent anti-parallel sliding that we see in our PRC1-Kif4A experiments.

We have added these new results to the main text and discussion (Figure 2) (Also see comment #6).

6) The relatively low affinity between KIF4 and PRC1 (~0.3 μM, cf. Kd~8 nM between PRC1 and anti-parallel microtubule bundle) raises the question of what percentages of KIF4 and PRC1 in the tagged and untagged overlaps are in the complex. This might not be important if there is no productive force generation by KIF4 molecules not complexed with PRC1. However, considering the presence of the second microtubule-binding site of the KIF4 tail (see above), it is important to clarify the states of the PRC1-KIF4 complex formation. Reduction of the sliding velocity by increased PRC1 and the reversal by additional KIF4A is consistent with a picture that PRC1 holds the anti-parallel overlap of MTs and KIF4A, which is not in a complex with PRC1, slides them. Please discuss.

We thank the reviewers for raising this interesting point. There are three possible sets of molecules bound to microtubules under these experimental conditions – PRC1 dimer, Kif4A dimer and PRC1-Kif4A complex. Here, we discuss the distribution of these molecules at the untagged and end-tagged regions of the antiparallel overlap.

At the untagged overlap: As discussed above, we find that the C-terminus tail of Kif4A does not directly bind microtubules (Figure 2). We have no evidence that Kif4A oligomerizes to drive sliding (both from gel-filtration as well as the inability to form microtubule bundles with Kif4A alone in ATP). So, the molecules within the untagged region of the antiparallel overlap that are responsible for sliding are likely to be PRC1-Kif4A complexes. However, as noted by the reviewer, the K_d_ for the PRC1-Kif4A affinity is 0.3 μM. Consistent with this, we have not been able to isolate stable PRC1-Kif4A complexes by gel filtration or visualize complexes by single-particle EM. However, PRC1-Kif4A complexes have a longer microtubule lifetime than either protein alone. Together, these data suggest that the complexes are undergoing dynamic dissociation and re-association on the microtubule lattice. When we increase Kif4A concentration, more PRC1 molecules will be in a sliding-competent complex and contribute to sliding. The cross-bridging and sliding by the dynamic complex is likely to play a key role in the observed length-dependent end-tagging observed in our experiments by further decoupling motor stepping from microtubule sliding (these points have been added to the Discussion section and Appendix sections of the main text).

At the end-tagged regions of the overlap: Analysis of fluorescence intensities suggests that at the time point when the transition to final overlap occurs, there is a 5-fold excess of Kif4A over PRC1 at end-tags. So, the end-tags are likely to be a mixture of PRC1-Kif4A complexes and Kif4A alone.

Our experiments suggest that end-tags composed of Kif4A alone can drive relative microtubule movement [Figures 2C-D]. Since end-tags are highly crowded regions of slowly dissociating protein, it may lead to a fraction of kinesins to adopt a one-head bound state (as is seen in EM experiments performed at high concentrations of kinesin dimers [Luduena, 2013; Yu, Garnham and Roll-Mecak, 2015]). In addition, the molecules at the very tip of the microtubule may also be predominantly in a one-head bound conformation. We propose that one-head bound kinesins can collectively drive end-on attachment and sliding of one microtubule over another. This is similar to a recent proposal for dynein driven relative-microtubule sliding to form minus-end asters [Gadadhar et al., 2017]. Interestingly, the association angle between two microtubules is observed to be low in these experiments. Whether motors at end-tags contribute to sliding in a PRC1 crosslinked antiparallel bundle (180 degrees) is unclear (and difficult to untangle given that the sliding and end-tag formation are linked phenomena).

Can such a mechanism also drive length-dependent sliding in the non-end-tagged overlap? We think this is unlikely for three reasons. First, in our control Kif4A experiments, sliding velocity is independent of the amount of protein at microtubule ends. Second, in the Kif4A experiments, we are unable to form aligned antiparallel bundles at both low and high ATP conditions. This result indicates that Kif4A does not cross-bridge microtubule in the canonical manner. Third, PRC1 crosslinks span an inter-microtubule distance that is in the 35 nm range. The likelihood of two kinesin motor domains connected by a 12 aa long neck-linker spanning that distance is unlikely.

For these reasons, we propose that in the untagged overlap, a cross-bridging PRC1-Kif4A complex drives sliding, whereas in the end-tagged region, sliding may be mediated by the collective action of single motor heads at the microtubule end/tip. However, the scaling of sliding velocity with overlap length only arises from molecules in the untagged overlap. We have now revised our mechanism figure (Figure 8) to reflect the new data.

7) Which fraction of KIF4-PRC1 is driving the sliding and what is causing the slow-down (during the phase-2). Is the decrease of velocity in phase-2 due to the decrease in the number of "untagged" KIF4-PRC1 or an increase in their density? Does friction play any role? Please discuss.

There are two fractions of molecules that can drive sliding: (i) crosslinking PRC1-Ki4A complexes and (ii) Kif4A molecules at the end-tag. However, our data suggest that the microtubule overlap length-dependent sliding only arises from sliding by molecules in the untagged overlap (Figure 4—figure supplement 1D-E).

As previously discussed (response to comment #2), our data suggest that the slowdown is likely due to an increase in density close to end-tags. The transition from sliding to stalling coincides with the end-tags on the moving and immobilized microtubules arriving at close proximity. At this high-density region, microtubule sliding would first slow down before complete stall is reached (Figures 4E-F). We noticed that in ~50% of the events (47/98; 0.2 nM PRC1 + 6 nM Kif4A-GFP), the transition from phase 1 to 2 occurs when the end-tags have nearly merged and our image analysis algorithm does not resolve the two end-tags (example: see kymograph in Figure 4E). We reasoned that if high protein concentration proximal to end-tags is the reason for slow-down, the protein density at the phase-1 to phase-2 transition should be similar under different experimental conditions. Kif4A-GFP density measurement and fluorescence line-scan analysis at the phase-1 to phase-2 transition time-points in two experimental conditions show that this is indeed the case (Figure 4—figure supplement 4A). Furthermore, comparison of the phase-1 to phase-2 transition-point intensity with the average end-tag intensity suggests that on average slowdown occurs when the intensity is >70% of the average end-tag (Figure 4—figure supplement 4B-C). These observations suggest that sliding slows down proximal to end-tags, when the untagged overlap is short and the motors encounter a high-density region, where stepping is inhibited.

While we cannot completely rule out the contribution of friction to the reduction in sliding velocity, the following lines of evidence suggests that it is unlikely to significantly impact the movement velocity in this system. First, previous measurements of frictional forces generated by PRC1 show that the magnitude of these forces is low [Forth et al., 2014]. Second, PRC1 increases processivity without decreasing Kif4A movement velocity suggesting that the drag forces generated by PRC1 are low [Bieling, Telley and Surrey, 2010]. Third, under these conditions, the total number of PRC1 molecules is less than the number of Kif4A molecules in the overlap (Figure 8—figure supplement 1), further arguing against frictional forces being the predominant reason for slow-down.

We have now included these points in the main text.

8) The description of the result from subsection “Characterization of relative microtubule sliding in the PRC1-Kif4A system” (density and intensity of Kif4A in the different overlap regions) is not very clear. What is meant with "retained"? What is meant by the final sentence of this part? Are these statements based on only two example kymographs (Figure 3E and Supplementary figure 1C-L). For Figure 3H it appears that the overlap keeps on increasing until the end of the experiment (unlike the event in sup 1C-L). Please explain more clearly.

Our word choice of ‘retained’was inspired by work from Stefan Diez’s group. However, we realize that this may be a confusing term in our paper and have removed it from the text.

We have analyzed more kymographs and included a panel of examples and quantification (Figure 4—figure supplement 2). These kymographs are representative of the data and the increase in density with overlap shrinkage is observed in all events we have analyzed. The initial increase in overlap intensity represents the equilibrium establishment phase. In Figure 4E in the current manuscript (Figure 3H in the previous submitted manuscript), the time is longer for 𝐼_𝑜𝑣𝑒𝑟𝑙𝑎𝑝_ to reach equilibrium for that particular kymograph. We have now also analyzed equilibrium data when 𝐼_𝑜𝑣𝑒𝑟𝑙𝑎𝑝_ is constant and these data also show scaling of sliding velocity with initial overlap length (Figure 4—figure supplement 3).

9) To what extend is the analysis of the end tag lengths (Figure 4AB) affected by the diffraction limited imaging used? How different are the end-tagging lengths on single microtubules and in microtubule overlaps? Please discuss.

We cannot make reliable length measurements when the end-tags are short (< 5 pixels) but we can measure intensity more accurately at these length scales. It is known that end-tag length and intensity are tightly correlated parameters [Subramanian et al., 2013]. Therefore, we have included both length and intensity data for all analyses requiring quantification of end-tags (Figure 4—figure supplement 1A-B). The method for measuring the end-tag lengths and intensity are in Materials and methods section.

10) In their theoretical model, the authors implicitly assume that one of the two PRC1 ends that binds the same microtubule with the one that KIF4 interacts with (i.e. the bottom one in Appendix Figure 1A) doesn't interfere with the MT sliding. It is not clear how this is justified. It would be more natural that one end of a PRC1 dimer (upper one in Appendix Figure 1A) works as a supporting point for productive sliding while the other end (bottom one) works as a drag against the stepping of KIF4. Please discuss and resolve the issue.

While PRC1 increases processivity of Kif4A, it does not alter motor velocity on single microtubules [Bieling, Telley and Surrey, 2010]. We have also previously observed that Kif4A movement proceeds at high velocities in the presence of PRC1 except at end-tags where motor stepping is inhibited (supplement in [Subramanian et al., 2013]). Further, it has been shown that the frictional force associated with PRC1microtubule interaction is ~2 orders of magnitude less than the typical forces generated by kinesin [Forth et al., 2014]. These observations are the basis for our assumption that the spectrin domain of PRC1 that shares the microtubule with the Kif4A motor domains does not generate significant drag force.

Cryo-EM structural analysis has shown that PRC1 and kinesin have partially overlapping tubulin-binding interface [Kellog et al., 2016]. This result suggests that at high-density PRC1 may sterically impede kinesin movement, as is observed at end-tags on single microtubules [Subramanian at el., 2013]. This is also consistent with our analysis of protein density and intensity at the phase-1 to -2 transition point (Figure 4—figure supplement 4 and response to comment #7). Therefore, while we cannot completely rule out the effects of frictional forces, we propose that steric hindrance to stepping is likely to be the predominant mechanism of slow-down and stall in the PRC1-Kif4A system.

We have modified the text in the Appendix clarifying these points and also revised the schematic that makes our theoretical model assumptions clear.

11) On one hand, modeling the PRC1-MT interaction as a slipping tether using the formulation by Grover et al., is an excellent idea. However, the model presented by the authors is not properly describing their experimental conditions. Their model is for a different situation in which KIF4 is tethered on the PRC1 bridging two microtubules (one of them is immobilized) and it drives the sliding of the third, non-immobilized microtubule. Thus, it is not surprising that there is a big inconsistency between the model calculation and the experimental measurements in the order of the sliding velocity. Please resolve.

We apologize for the lack of clarity. In our experiments (except Figure 7) we only examine pairs of microtubules and our model is describing the same molecular configuration (revised Figure 1 of the Appendix) where Kif4A is linked to the PRC1 molecule bridging two microtubules (one of them is immobilized and the other is the moving microtubule; we show just one protofilament of each microtubule to simplify the schematic). Our goal was to determine if modeling the PRC1-MT interaction as a slipping tether using the formulation by Grover et al., could give rise to overlap length-dependent sliding. We find that even this simple model can recapitulate length dependent sliding with velocities that are roughly in the same range as in our experiments. While other factors, such as steric hindrance, occupancy, and PRC1-Kif4A dissociation and reassociation kinetics will impact the magnitude of the sliding velocity and the extent to which velocity scales with overlap length, the minimal mechanism is sufficient to recapitulate the overall trend observed in our experiments. We therefore propose that this is a possible mechanism for initial overlap-length dependent sliding that is consistent with our experimental observations.

12) On the other hand: Is it really warranted to use a similar model to diffusion (and drag) in a membrane for diffusion of the PRC1 complexes over the MT? PRC1 binds to the MT, most likely with 8 nm periodicity, with relatively large barriers / wells; in the membrane stuff is much more continuous. Would in the current situation such a view not be too simplistic, e.g. not taking into account 'non-linear' / 'out-of-equilibrium' effects due to motor action (i.e. increased loads on the motor which could result in changes in motor action (velocity, release) including a non-linear scaling of friction with motor number (see Lansky et al., 2015).

We agree that the model we consider here is a simple and minimal one. However, the main purpose of the computational work was to determine if modeling the PRC1-MT interaction as a slipping tether could give rise to initial microtubule overlap length-dependent sliding (phase-1) as observed in our experiments. This is especially interesting as current models for reduction of sliding velocity in shrinking overlaps (e.g. Ase1-Ncd and HSET systems) do not apply to the PRC1-Kif4A module [also discussed in response to summary and comments #2].

Currently, we do not have experimental data to model other details of the mechanism. For example, the microtubule binding interactions of PRC1 are mediated by its spectrin domain and an unstructured C-terminus domain (~150 aa) with three stretches of positively charged residues. It is proposed that the unstructured domain in PRC1 interacts with the negatively charged and unstructured ‘tail’ of tubulin. However, the distances over which they interact, periodicity and barriers, occupancy and how these are altered when PRC1 is in a complex with Kif4A are all unknown. Similarly, there are no force measurements that have been performed on this system yet (note: non-linear scaling of frictional forces if present in this system can reduce the magnitude of the sliding velocity but it cannot give rise to the scaling of sliding velocity with overlap as observed in our experiments, which is the focus of this modeling effort). We think that in the future more extensive modeling together with biophysical measurements will reveal other aspects of this system, but they are beyond the scope of this manuscript.

In addition, while certain details of the molecular interactions in this system are unknown, there are some aspects that are known and feed into the model. These include low frictional drag forces generated by PRC1, lack of entropic forces of significant magnitude in this system, diffusive nature of PRC1 interaction, and the moderate PRC1-Kif4A binding affinity. While we only model diffusive tether as the decoupling mechanism (since we have the most experimental data to support this model), there are likely to be others such as dissociation of the PRC1-Kif4A complex that reduce the coupling between motor stepping and microtubule sliding. However, even this minimal model captures the trend and magnitude of scaling observed in our experiments.

13) According to the authors' formula in the Appendix, 𝑣_MT_ is hyperbolic against a dimensionless value x=a∙d∙l/L_MT_ ("Michaels-Menten" type, passing origin and approaching max value = 𝑣_step_). A characteristic parameter is f∙δ, which corresponds to x which gives a half maximal 𝑣_MT_ (equivalent to K_M_ for "Michaels-Menten"). In a regime where x is below this, the 𝑣_MT_~ x relationship becomes nearly linear. However, calculations with the values in Table A1 result in f∙δ = 1.72 x 10^-5. This is too small, and it is impossible to make x smaller than this by any realistic combinations of a, d, l and L_MT_. In other words, 𝑣_MT_ is almost equal to 𝑣_step_ irrespective of l, the lengths of overlap. Please explain how the curves in Appendix Figure 1 were drawn. The actual values of a, d and D used should be presented.

In our theoretical model calculations to produce Appendix Figure 1 𝑘_𝐵_𝑇 was converted to units of N∙m units from units of J which is 𝑘_𝐵_𝑇~ 4 pN∙nm. The 𝑘_𝐵_ value in the Table A1 is in units of J/K. We apologize for the confusion this caused. We have replaced this 𝑘_𝐵_ value with consistent units in Table A1. In Appendix Figure 1, the actual values of 𝐿_𝑀𝑇_, 𝑎, 𝑑, and 𝐷 are the following:

Figure 1B: for 𝐿_𝑀𝑇_ = 2, 4, and 6 μm: 𝑎 = 40%; 𝑑 = 13; and 𝐷 = 2000 nm^2^/s

Figure 1C; 𝐷 = 200 and 2000 nm^2^/s: 𝐿_𝑀𝑇_ = 6 μm; 𝑎 = 40%; and 𝑑 = 13

Figure 1D; 𝑎 = 10 and 40%: 𝐿_𝑀𝑇_ = 6 μm; 𝑑 = 13 and 𝐷 = 2000 nm^2^/s

These values have also been updated in the legend of Appendix Figure 1 for each subplot.

*14) The modeling should be connected closer to the experiments. For example, a∙d is essentially the line density of PRC1-KIF4A in the overlap and thus should be measurable. Then, the actual D should be able to be determined with actual x and* 𝑣_𝑀𝑇_
*measurements using the authors' formula and can be compared with reported values. Moreover, it would be helpful to indicate what trends (and numbers) belong to the experimentally tested parameter space. The simulations (at least their results) should be discussed more prominently in the main text /Discussion section. Please add this information.*

We thank the reviewers for raising this point. We have extracted 𝑎 ∙ 𝑑 from the untagged overlap density (intensity/length) from our experimental data. Using the 𝑣_𝑀𝑇_ and x values, and the formula from Appendix, Eq. 5, the 𝐷 value is determined to be ~2000 nm^2^/s (N = 10; 0.2 nM GFPPRC1 + 6 nM Kif4A). In comparison to reported values, our 𝐷 value is consistent with Bieling et al., (𝐷 = 2900 nm^2^/s) for PRC1 within microtubule overlaps.

Our modeling qualitatively recapitulates the extent of scaling of sliding velocity with initial overlap length (Appendix Figure 1; note: magnitudes are in the experimental range assuming values for occupancy, which we cannot determine from our intensity data as discussed in response #1). The model predicts that the scaling of sliding velocity with initial overlap length depends strongly on motor occupancy and PRC1-diffusivity, and weakly on MT length. Consistent with these trends, our experimental data (Appendix Figure 1; gray circles; 0.2 nM PRC1 + 6 nM Kif4A-GFP), show a weak dependence of the length of the moving microtubule and a stronger dependence of sliding velocity on initial overlap length at higher Kif4A-PRC1 ratios where we would expect a greater percent of complexes in the overlap contributing to sliding (Figure 4A-B).

We have added a discussion of the modeling results to the main text. Since the Appendix will immediately follow the main text in the final publication, we prefer to leave the details of the modeling out of the Discussion section.

15) Biological significance: It is unclear whether the situations studied in this work actually occur in the cell. Although the end-tagging of astral microtubules near spindle poles by PRC1 was demonstrated in Subramanian et al., 2013, it remains unclear whether KIF4A takes part in these tags. Even so, it is unclear whether there is a cellular situation in which microtubules are first tagged (and stabilized) with PRC1 and KIF4A and then bundled. In general, PRC1 localizes on the metaphase spindle although weakly and diffusely while KIF4 is associated with chromosomes before anaphase onset. PRC1 can form midzone bundles without KIF4. Please discuss.

In recent years, the multiple biochemical activities that a single kinesin can perform has been revealed through in vitro characterization of the motor. For example, Kip3p, best known as a regulator of microtubule dynamics was shown to crosslink and slide microtubules to organize microtubule arrays [Su et al., 2013]. Similarly, Eg5, which is best known for crosslinking and sliding, has recently been shown to also regulate microtubule dynamics [Chen and Hancock, 2015]. The role of the PRC1-Kif4A module in regulating microtubule dynamics was first revealed through beautiful reconstitution studies from the Surrey lab [Bieling, Telley and Surrey, 2010]. However, while microtubule sliding was also briefly mentioned in their paper, this activity remains completely uncharacterized. The goal of this study is to address this poorly understood aspect of PRC1-Kif4A function. To test whether sliding contributes to midzone length control, we would need a Kif4A mutant that retains PRC1 binding and sliding and only perturbs microtubule dynamics. Unfortunately, there is currently little understanding of how Kif4A regulates microtubule dynamics at a structural level. The structural investigation and cell biological assays are beyond the scope of this work.

While we do not know if microtubule sliding by Kif4A contributes to the midzone organization, several properties of this system are relevant to cell biological scenarios both during mitosis in eukaryotes and in interphase cells of yeast and plant cells. For example, kinesin accumulation at microtubule ends is observed for several motors [Subramanian et al., 2013; Su et al., 2013; Varga et al., 2006; Leduc et al., 2012; Vitre et al., 2014]. While we only looked at PRC1 localization in monopolar and bipolar spindles, a recent work from Tim Mitchison’s group shows that both PRC1 and Kif4A molecules are at the plus ends of asters in *Xenopus* egg extracts [Nguyen, Field and Mitchison, 2018]. The ends of microtubules are difficult to image in the midzone of a human cell. However, work from Ted Salmon’s group suggests that when the midzone assembly is perturbed, end-tags are observed even in the central microtubule bundle [Shannon et al., 2005]. This suggests that transport and end-accumulation may be features of the midzone and perhaps contribute to the stability and close alignment of microtubule plus-ends in this antiparallel microtubule array.

Whether sliding of PRC1-crosslinked microtubules contributes to the midzone organization in mammalian cells remains unknown at this time. Recent work from Iva Tolic’s group shows that sliding of PRC1-crosslinked interkinetochore bridges contributes to chromosome segregation in anaphase [Vukusic et al., 2017]. At least one of the motors involved in central spindlin, which also interacts with PRC1. Further, yeast, mammalian and plant cells all have PRC1-crosslinked microtubule arrays that have associated motors with proposed sliding activities [Subramanian and Kapoor, 2012]. Initial overlap length-dependent sliding could be advantageous in ensuring that microtubules of different lengths arrive at similar rates to the plus-ends of the template microtubule within arrays. Therefore, the features uncovered here are likely to inform our models of how the collective activities of these proteins regulate the organization of various cellular microtubule arrays.

Finally, the reconstitution described here, examining microtubule sliding by a pair of interacting motor and non-motor crosslinking protein, reveals unique emergent properties that have not been reported previously. We expect that these findings will inform future models of microtubule self-organization.

We have now added some of the key points from this response to the Discussion section of the main text.

[Editors' note: further revisions were requested prior to acceptance, as described below.]

The involved editors as well as the reviewers acknowledge that you did a great job of seriously considering the earlier comments, including the performance of additional experiments. The conclusions are now much more solid and the context with previous studies and other systems is discussed much more clearly, highlighting why this study is important and exciting. The manuscript has tremendously gained by the revision and it is felt that the work in general is very well suited for eLife.However, there are some remaining issues that need to be addressed before your manuscript can possibly be accepted for publication:1) Please have a look at the following (conclusion of the Figure 4, subsection “Examining the time-dependent changes during microtubule sliding in the PRC1-Kif4A system”): "…velocity of microtubule sliding.… is determined by.…. the total number of sliding competent molecules in the untagged overlap." Velocity increases with increasing (untagged) motor number (Figure 4A-D) and "The microtubule movement can subsequently proceed at a *constant velocity*, even when the overlap shrinks,…" (i.e. when the untagged *motor number decreases* – phase 1 in the Figure 4F and Figure 4—figure supplement 2) "…possibly through increasing the density of motor molecules during relative sliding." This would mean that increasing the motor density should compensate for decreasing the motor number to keep the velocity constant. That is, velocity should increase with increasing motor density.However: (i) How an increased motor density would result in an increased sliding velocity is intriguing and the authors should probably comment on this. (ii) In contradiction with their statement, the authors show that (at least in some concentration regime) this is not the case (Figure 4—figure supplement 1F).

We realize that this statement in the text: "…possibly through increasing the density of motor molecules during relative sliding", may be confusing and we have re-written this subsection “Examining the time-dependent changes during microtubule sliding in the PRC1-Kif4A system”.

Our data show that under a given experimental condition, sliding velocity scales with initial overlap length and intensity. Thus, sliding velocity scales with motor number in the initial overlap. This suggests that for overlaps of the same initial length, increasing the motor density will increase the total number of sliding-competent motors, which in turn should result in higher sliding velocity. In Figure 4—figure supplement 1F, we are plotting the average density during phase-1 movement from events that have different initial overlap lengths. We do not have sufficient events that have the same initial overlap length but different densities under the same experimental condition to perform a correlation analysis.

As an alternative, we re-examined events from the data shown in Figure 3—figure supplement 1, where we plot the average sliding velocity in experiments with 1 nM GFP-PRC1 + 6 nM Kif4A and 1 nM GFP-PRC1 + 12 nM Kif4A. We expect that the density of Kif4A will be greater at the higher motor concentration. We replotted the sliding velocity (phase-1) at comparable initial overlap lengths under these two conditions and find that the sliding velocity is higher at 12 nM Kif4A concentration (Author response image 3). Intensity analysis of GFP-PRC1 shows that the increase in sliding velocity at 12 nM Kif4A is not due to a decrease in the density of PRC1 molecules at the antiparallel overlap under these conditions (Author response image 3).

These data suggest that at comparable initial overlap length and comparable PRC1 levels, increasing the Kif4A concentration, which would increase the total number of motor molecules in the initial overlap (and consequently the density in overlaps of similar lengths), results in an increase in sliding velocity.

**Author response image 3. respfig3:** Sliding velocity as a function of initial overlap length and untagged overlap density. A) Sliding velocity as a function of initial overlap length for two bin sizes. Assay condition: 1 nM GFP-PRC1 + 6 nM Kif4A (gray; 500-1500 nm: N = 13, 1500-2500 nm: N = 18) and 1 nM GFP-PRC1 + 12 nM Kif4A (blue; 500-1500 nm: N = 13, 1500-2500 nm: N = 9) **B**) Histogram of the untagged overlap density of GFP-PRC1. Assay condition: 1 nM GFP-PRC1 + 6 nM Kif4A (gray; N = 38) and 1 nM GFP-PRC1 + 12 nM Kif4A (blue; N = 18).

2) Connected to (1): An essential statement of the paper is that the sliding velocity scales with the initial overlap length. However, Figure 4E-H shows that the sliding velocity stays constant for a shrinking overlap lengths (called phase 1). How is the "initial overlap length" defined here? What means "initial"? Most surprisingly, the overlap intensity increases during this phase. How is that explained? Is there equilibrium in binding achieved before?

In the PRC1-Kif4A experiments connected to Figure 1 of the manuscript, we first set up pairs of PRC1-crosslinked antiparallel microtubules as described in the methods. We then initiate sliding by flowing in a mixture of PRC1, Kif4A and ATP. The initial overlap length is defined as the microtubule overlap length at the first time-point we acquire (𝑡 = 0; example first panel in Figure 1B). We have now emphasized the definition of “initial overlap length” in the text and Materials and methods section, subsection “Image Analysis”).

The initial increase in intensity (note: the intensity reflects both the crosslinking motors that contribute to sliding and passenger molecules that do not contribute to sliding) is due to the establishment of chemical equilibrium and this is observed in all the events we have looked at. As described in our previous response to reviewers (#8) and the submitted revision, we re-analyzed a subset of events in which we could measure sliding velocity in phase-1 after equilibrium is established (i.e. total overlap intensity levels are constant; I_overlap_). The results are identical to our findings from analyzing the complete time course of phase-1 (Figure 4—figure supplement 3).

3) With regard to the interaction between the C-terminal tail of KIF4A and microtubules it is stated that the interaction between the KIF4A tail and microtubules are not strong. However, the PAGE image in Figure 2 clearly shows there is some interaction between them. The signals of the bands that correspond to KIF4A (but don't appear in 'PRC1' lanes) in the precipitates (KIF4A(C-term) 'P') increase with the increasing amount of microtubules. This indicates a weak but significant interaction between KIF4A C-tail and microtubules. The authors' statements such as "We observed no significant microtubule-association of this domain" (subsection “Molecular determinants of the sliding and cross-bridging in the PRC1-Kif4A system”) or "the C-terminus of Kif4A does not directly bind microtubules" are thus not true.This interaction is indeed 'very' weak as a MAP. The dissociation constant might be at the orders of 100 µM or 1 mM. However, it should be kept in mind that this domain is not floating alone in solution but is part of a kinesin-like motor protein, which strongly accumulates at the plus-ends of microtubules. The local concentration of the domain can be the order of 1~10 mM, which seems to be comparable with the weak but significant interaction detected in Figure 2A.Similarly, hydrodynamics data at the protein concentration of 5 µM might not be strong enough to exclude the possibility that Kif4A might form oligomers at the crowded condition. The second (very weak) binding site on the C-terminal tail or oligomerization seems to be more plausible as an explanation of the MT-sliding by Kif4A alone. At least the data are not strong enough to exclude these possibilities. Please discuss all of the above.

We have quantified the SDS-PAGE gels from the co-sedimentation experiments. The data does not show tubulin concentration dependent binding (Author response image 4; black and orange data points), and therefore we cannot conclude that the C-terminus of Kif4A is a bona-fide microtubule-binding domain from our data. In our experience, most proteins, including BSA, pellet at high concentrations of microtubules to some extent depending on the pH and ionic strength of the assay (Author response image 4; purple and red data points). These quantifications are now included in the figure legend of Figure 2 of the main text. However, we agree with the reviewer that we cannot rule out that this domain is an extremely weak MAP with K_d_ on the order of 100 µM-1 mM. Whether such low affinity interactions can sustain sliding (due to the high off-rates of the motor from the protofilament) when the motor is concentrated at the microtubule end is currently unknown. Furthermore, the C-terminus of Kif4A also binds PRC1 (K_d_ = 0.3 μM). Therefore, it is possible that the interaction between the Kif4A-C term and microtubules, if it does occur, is further reduced in the presence of PRC1. We agree with the reviewers that we cannot completely exclude the possibility that some small fraction of motors may form sliding-competent oligomers at the ends of microtubules due to high local concentrations. However, this is not easily to test experimentally.

Overall, the molecular mechanism of Kif4A-end-tag mediated sliding, whether it is through weak Kif4A-Kif4A oligomerization or low-affinity Kif4A-microtubule interaction or through single unattached motor domains at the highly dense end-tags, appears to be different from other crosslinking kinesins. We have revised the text to mention these additional possibilities (subsection “Mechanism of overlap-length dependent sliding by PRC1 and Kif4A”). However, since end-tags alone are not sufficient to generate overlap length-dependent sliding in the PRC1-Kif4A experiments (also see response #6), the specific mechanism by which Kif4A end-tags drive sliding in the absence of PRC1 does not impact the main conclusions from this study and will be examined in the future.

**Author response image 4. respfig4:** Quantitative analysis of SDS-PAGE gels from two co-sedimentation assays. The percentage of Kif4A (black and orange) and BSA (red and purple) in the pellet is plotted against tubulin concentration. The curves labeled bound-1 are connected to the gel in Figure 2 of the main text. Inset: Zoomed-in view of the plot.

4) The response to comment 13 is unsatisfactory. The difference of units doesn't matter if the calculation is performed with physical quantities (numbers + units).The characteristic parameter (the initial overlap that gives the half maximum velocity, analogous of K_M_ in the Michaelis-Menten kinetics, here called λ) λ should be at the order of µm or bigger. However, when repeating the calculation with the values provided in the revision, the λ is calculated to be at the order of picometer. Thus, what the authors' theory predicts is that the sliding velocity is independent of the initial overlap length. The theoretical curves in Appendix Figure 1 appear inconsistent with the authors' theory and the parameters provided. Please check.5) In the calculation, d = 13 and a = 0.4 is assumed – meaning that there should be about 600 molecules active in a 1 micron long MT overlap? Is that a reasonable assumption? Can't an upper bound of the motor number be (roughly) estimated from the fluorescence intensity?

We are extremely thankful to the reviewers for discovering this error during the peer review process. We discovered that there was a mistake (in converting units) when extracting the diffusion constant as described in the Appendix of the manuscript. A summary of new analyses and outcomes is as follows.

(1) The recalculated diffusion constant is 9x10^7^ – 3x10^8^ nm^2^/s (for 1-8% protein occupancy estimated from experimental data; see below). These values are ~10^5^ fold greater than the reported diffusion constant of PRC1 within antiparallel overlaps (2900 nm^2^/s; Bieling et al., 2010). As pointed out in comment # 4, the published values do not quantitatively explain the overlap-length dependent rate of sliding observed in our experiments. Therefore, we need to consider mechanisms other than diffusive anchorage of PRC1 that can decouple motor stepping from microtubule sliding. At the molecular level, we can imagine at least two other possibilities: (i) dissociation of the PRC1-Kif4A complex within overlaps and (ii) increased dissociation rate of PRC1 from the moving microtubule due to motor stepping. However, we feel that determining the precise molecular mechanism by which motor stepping is decoupled from microtubule sliding is beyond the scope of this paper.

Based on the above findings, we took a step back and decided to estimate the duty ratio of the PRC1-Kif4A molecules instead of assuming specific mechanisms. These calculations, which are summarized in Author response image 5, provide an approximate estimate of the duty ratio (approximate because (a) we do not reach saturation sliding velocity in our experiments and (b) we are estimating number of sliding-competent molecules from fluorescence data (now calculated experimentally – see next point)). We find that the estimated values of duty ratio are less than 1, and in the same range as those published for myosin [Uyeda, Kron and Spudich, 1990], β dynein [Imafuku, Toyoshima and Tawada, 1997], 22S dynein [Hamasaki et al., 1995], and NcKin3 [Adio et al., 2006], which exhibit filament length/motor-number-dependent movement velocities. This suggests to us that while the precise molecular mechanism underlying the scaling of sliding velocities with initial microtubule overlap-length is currently unknown, it is reasonable to think about microtubule sliding by the PRC1-Kif4A complex as microtubule movement driven by an ensemble of low duty-ratio motors. We have altered the Discussion section to focus on duty ratio and the potential molecular mechanisms that could result in a reduction of the duty ratio and give rise to length-dependent sliding in the PRC1-Kif4A system.

**Author response image 5. respfig5:** A) Calculation of the duty ratio of the PRC1-Kif4A molecules. B) Estimation of the number of molecules/µm from experimental fluorescence intensity measurements. C) The fitting of Eq. 1. (red line) to the microtubule sliding velocity as a function of initial microtubule overlap length data (Assay condition: 0.2 nMPRC1 + 6 nM Kif4A-GFP (gray circles; N = 84).

We thank the reviewer for raising the point of occupancy number in a microtubule overlap. We have now measured intensities of single Kif4A-dimers and used it to estimate the maximum protein occupancy in microtubule overlaps (see Author response image 5). The average occupancy (𝑎) is estimated to be 1% if PRC1 molecules can crosslink to all 13 microtubule protofilaments of the moving microtubule and 10% assuming that effective crosslinks are only formed with one protofilament. Considering the molecular structure of PRC1, the values are likely to be in the 1-10% range. These values were used to calculate the duty ratio (Author response image 5).

6) With regard to the contribution of the end tags on sliding: Kif4A alone can form an end tag, which drives movement of a non-immobilized microtubule along an immobilized one. It is not clear why the similar end-tag doesn't contribute much to the MT sliding in the Kif4A-PRC1 regime. How fast is the movement driven by Kif4A alone end tags in the experiments represented by Figure 2C and D (can't be currently estimated because scale bars are missing in these panels)?

As noted by the reviewers, we find that Kif4A end-tags alone can drive the movement of one microtubule over another (Figure 2). The velocity data for the Kif4A-alone driven sliding was presented in Figure 4—figure supplement 1Q inset (v = 75 ± 25 nm/s). Scale bars have now been added to Figure 2C-D. Indeed, it is possible that end-tags contribute to antiparallel microtubule sliding in the PRC1/Kif4A system, and we indicated this as ‘potentially sliding’ in the schematic in Figure 8. However, the extent to which PRC1-Kif4A end-tags contribute to sliding is unclear at this time due to the following reasons:

Sliding velocity in the PRC1-Kif4A experiments scale with untagged overlap length and not the end-tag length (Figure 4—figure supplements 1A, B, D, E). We do not observe a correlation between end-tag intensity and sliding velocity in Kif4A-alone experiments (Figure 4—figure supplement 1Q). Together, these data suggest that molecules in the untagged overlap and not those at the end-tag drive length-dependent antiparallel sliding in the PRC1-Kif4A experiments.

We can compare the sliding velocities in the standard PRC1-Kif4A sliding experiments (Figure. 1 of main text) with the velocities in the ‘wash-out’ experiments (Figure Author response image 1; Author response image 6). Briefly, in the wash-out experiments that were performed during the previous round of revision, we first establish microtubule bundles with proteins and ADP (0.2 nM GFP-PRC1 + 6 nM Kif4A and 2 mM ADP). Next, the solution protein was washed-out and exchanged with buffer containing ATP (no additional protein included). Under these conditions, when the protein concentration in the solution is low, it is observed that end-tags are re-established through depletion of PRC1 and Kif4A from the untagged overlap, and sliding is simultaneously reinitiated. Therefore, the washout experiments present a scenario in which end-tags are robustly established but the amount of protein in the untagged overlap is low.

If end-tags are sufficient to drive antiparallel sliding and molecules in the untagged overlap do not contribute to velocity, then the sliding velocities in the wash-out experiment would be in the same range as those seen in Kif4A sliding experiments. Instead, we find that the average velocity in the wash-out experiments is lower (v = 15 ± 5 nm/s) suggesting that motors in the untagged overlap play a significant role in microtubule sliding in the PRC1-Kif4A system (Author response image 6). The predominant initial interaction angle between microtubules in the Kif4A-alone experiments is between 0-30° (Figure 2). It is possible that there are geometrical constraints that inhibit end-tag mediated sliding when microtubules are anti-parallel.

In summary, it is possible that Kif4A end-tags can slide antiparallel microtubules but this mechanism alone is neither sufficient to generate movements at the observed velocities nor give rise to overlap-length dependent sliding.

**Author response image 6. respfig6:** Average sliding velocity for PRC1-Kif4A no wash-out experiment (0.2 nM PRC1 + 6 nM Kif4A-GFP, N = 84; 60 ± 17nm/s) and wash-out experiment (0.2 nM PRC1 + 6 nM Kif4A-GFP + 2 mM ADP, N = 22; 15 ± 5 nm/s).

[Editors' note: further revisions were requested prior to acceptance, as described below.]

Thank you for your second resubmission of your work entitled "Geometry of antiparallel microtubule bundles regulates relative sliding and stalling by PRC1 and Kif4A" for further consideration at eLife. Your revised article has been evaluated by Anna Akhmanova (Senior Editor), a Reviewing Editor, and three reviewers.The involved editors as well as the reviewers acknowledge that you did a great job in addressing the points raised. Removing the earlier modeling part and replacing it by a kind of "duty ratio" discussion makes sense and is a nice way to extract quantitative information out of the data. While this current description is admittedly not as advanced / informative as a real model (attempted in the last version of the manuscript) it nevertheless provides useful mechanistic insight. Given the enormous amount of very high quality experimental data, the taken approach is regarded fine for this paper. Future work could go into a more advanced model. Hence, the manuscript is now in principle regarded suitable for publication in eLife.There is one remaining point that the reviewers and editors find of crucial importance before potential acceptance of the paper: The usage of the term/concept "duty ratio" does not seem to be fully appropriate in the presented context. The traditional/authentic "duty ratio" is about the temporal fraction of the crossbridge cycle (= ATPase cycle) of a single motor head in which it is attached to the filament and makes its working stroke. In contrast, the situations the authors imagine are (i) dissociation of the PRC-Kif4A complex from the microtubule, and (ii) slippage of PRC1 on the MT. These will influence the fraction of ATPase cycles that actually result in the sliding of the non-immobilized microtubule, i.e., the fraction of the productive stepping by Kif4A. What the authors call "duty ratio, f", is a mixture of the authentic duty ratio (as to the crossbridge/ATPase cycle) and the effect of the futile cycles. It is not appropriate to skip these details and call the parameter simply "duty ratio". A better term to describe the scenario may be "sliding efficiency". In any case, the authors should explicitly mention that they mean something slightly, but substantially, different than what "duty ratio" has been used for before.In other words, both the authentic duty ratio and the fraction of productive stepping would influence the sliding velocity in a similar way, following the same form of a mathematical formula (2) in subsection “Mechanism of overlap-length dependent sliding by PRC1 and Kif4A”, as a first approximation. However, their meanings are quite different. A low duty ratio motor can still be highly energy efficient (like dynein). On the other hand, futile sliding simply wastes energy of ATP hydrolysis as a slippery between PRC1 and MTs or a dissociation between PRC1 and KIF4A. Along these lines: Is the low "duty ratio" of the PRC1-Kif4A complex in MT sliding consistent with its highly processive motility along a MT? A quantitative argument is necessary as to the difference in the loads on the PRC1-Kif4A complex between the two conditions; the MT sliding and the single particle motility. Statements like "…microtubule sliding by the PRC1-Kif4A complex can be considered as microtubule movement driven by an ensemble of low duty-ratio motors,.… " need to be revised accordingly.

We fully agree that the usage of the term ‘duty ratio’ in the text is not completely accurate. We thank the reviewers for suggesting ‘sliding efficiency’ and have re-named and clarified this parameter in the Discussion section of the text (subsection “Mechanism of overlap-length dependent sliding by PRC1 and Kif4A”). We have also distinguished the difference between the authentic duty ratio and sliding efficiency in the text.

The highly processive movement of Kif4A on single immobilized microtubules in the presence of PRC1 arises from an increase in the lifetime of Kif4A motors under these conditions. At a molecular level, this can be achieved through transient interactions of PRC1 and Kif4A on the immobilized microtubule. In contrast, for relative sliding, crosslinks between two microtubules need to be formed by PRC1-Kif4A complexes such that motor stepping can be translated to microtubule displacement (example: Kif4A dissociation from one PRC1 molecule and reassociation with a neighboring PRC1 will increase its lifetime but not lead to microtubule displacement). Such a difference in the molecular mechanisms between stepping and sliding by the PRC1-Kif4A complex can explain how a motor that is highly processive on a single microtubule is characterized by low sliding efficiency.

Additional references:

Chen, Y. and W.O. Hancock. Kinesin-5 is a microtubule polymerase. Nat Commun, 2015. 6: p. 8160.

Varga, V., J. Helenius, K. Tanaka, A.A. Hyman, T.U. Tanaka, and J. Howard. Yeast kinesin-8 depolymerizes microtubules in a length-dependent manner. Nat Cell Biol, 2006. 8(9): p. 957-62.